# Multi-cohort and longitudinal Bayesian clustering study of stage and subtype in Alzheimer's disease

Konstantinos Poulakis [1] ✉, Joana B. Pereira[1,2], J.-Sebastian Muehlboeck[1], Lars-Olof Wahlund[1], Örjan Smedby[3], Giovanni Volpe [4], Colin L. Masters[5], David Ames[6,7], Yoshiki Niimi[8], Takeshi Iwatsubo[8], Daniel Ferreira [1,9], Eric Westman [1,10], Japanese Alzheimer's Disease Neuroimaging Initiative* & Australian Imaging, Biomarkers and Lifestyle study*

Understanding Alzheimer's disease (AD) heterogeneity is important for understanding the underlying pathophysiological mechanisms of AD. However, AD atrophy subtypes may reflect different disease stages or biologically distinct subtypes. Here we use longitudinal magnetic resonance imaging data (891 participants with AD dementia, 305 healthy control participants) from four international cohorts, and longitudinal clustering to estimate differential atrophy trajectories from the age of clinical disease onset. Our findings (in amyloid-β positive AD patients) show five distinct longitudinal patterns of atrophy with different demographical and cognitive characteristics. Some previously reported atrophy subtypes may reflect disease stages rather than distinct subtypes. The heterogeneity in atrophy rates and cognitive decline within the five longitudinal atrophy patterns, potentially expresses a complex combination of protective/risk factors and concomitant non-AD pathologies. By alternating between the cross-sectional and longitudinal understanding of AD subtypes these analyses may allow better understanding of disease heterogeneity.

Brain atrophy in Alzheimer's disease (AD) is associated with cognitive decline and the topological spread of neurofibrillary tangles (NFT)[1]. Neuropathological[2–4] and in vivo neuroimaging[5,6] studies challenge the hypothesis of AD as a single entity, supporting the hypothesis of AD as a heterogeneous disease. It was recently suggested that the heterogeneity in AD can be explained using two main dimensions, severity and typicality, which emerge in the form of various biomarker and clinical expressions[7]. Four AD subtypes are reported in the literature based on regional atrophy and/or NFT spread: typical, hippocampal sparing, limbic predominant[7,8], and minimal atrophy subtypes. However, the most urgent questions are whether the observed heterogeneity reflects different disease stages or distinct subtypes, and if these subtypes finally converge at advanced stages of the disease[7].

[1]Division of Clinical Geriatrics, Department of Neurobiology, Care Sciences and Society, Karolinska Institutet, Stockholm, Sweden. [2]Clinical Memory Research Unit, Department of Clinical Sciences, Lund University, Malmo, Sweden. [3]Department of Biomedical Engineering and Health Systems (MTH), KTH Royal Institute of Technology, Stockholm, Sweden. [4]Department of Physics, University of Gothenburg, Gothenburg, Sweden. [5]The Florey Institute of Neuroscience and Mental Health, The University of Melbourne, Victoria, Australia. [6]Academic Unit for Psychiatry of Old Age, St George's Hospital, University of Melbourne, Melbourne, Victoria, Australia. [7]National Ageing Research Institute, Parkville, Victoria, Australia. [8]Unit for Early and Exploratory Clinical Development, The University of Tokyo Hospital, Tokyo, Japan. [9]Department of Radiology, Mayo Clinic, Rochester, MN, USA. [10]Department of Neuroimaging, Centre for Neuroimaging Sciences, Institute of Psychiatry, Psychology and Neuroscience, King's College London, London, UK. *Lists of authors and their affiliations appear at the end of the paper. ✉e-mail: konstantinos.poulakis@ki.se

Advances in biomarker research, data collection, and computational methods, have substantially enhanced our ability to study the heterogeneity in different diseases[9]. These computational methods unite various in vivo pathophysiological markers to model disease heterogeneity. Research on classification of AD patients into meaningful groups with neuropathological[4], neuroimaging[8,10], clinical[11], and biochemical[12] biomarkers has shed light on the heterogeneity underlying the clinical AD diagnosis. However, current findings are based on cross-sectional analyses, which increase the chance that identified patterns reflect patient groups observed in different disease stages rather than distinct disease subtypes. A recent study modeled subtype biomarker trajectories in vivo from cross-sectional imaging datasets to implicitly infer disease stages[13]. That is a first step towards assessing and accounting for disease staging. However, we cannot exclude the chance that the identified patterns may still reflect different disease stages, since longitudinal information was not used for clustering, only for characterizing subtypes post hoc. This assumption is partially confirmed in models with various biomarker types (increased disease specificity) but remains unrealistic when a well-defined timescale of events for each patient is not in place. Recent reviews that presented the current approaches for identifying subtypes in heterogeneous diseases[9] and summarized the existing AD subtypes in the literature, point out important data and methodological limitations that need to be overcome to reach a better understanding of the heterogeneity in AD[7,8,14]. According to their conclusions, the field is lacking longitudinal AD subtyping based on a clear timescale (i.e., age at measurement, age at disease onset) in order to disentangle disease stages from disease subtypes.

In this study, we aimed to assess whether heterogeneity in AD's brain atrophy patterns results from observing patients at different disease stages or reflects distinct subtypes with specific atrophy and cognitive trajectories. Longitudinal data were modeled with a longitudinal Bayesian clustering framework[15] over 8 years from the clinical disease onset (a clear timescale) to assess disease staging and heterogeneity simultaneously (previous studies used only cross-sectional data). This is a significant step towards the discovery of differential atrophy trajectories in AD, using structural magnetic resonance imaging (MRI) data from four international multi-center cohorts from four continents. Only amyloid-positive AD patients were included to increase diagnostic specificity (discovery dataset). In addition, with our approach, we could assess whether atrophy subtypes[7,8] converge during the disease course, a vital step towards understanding the heterogeneity in AD. Frequency predictions of the discovered atrophy patterns were performed in an external validation dataset to assess the ability of our model to classify new patients with one or two MRI timepoints available. Finally, we assessed between and within subtype differences in cognitive decline and relevant disease modifiers such as APOE genotype, education, and premorbid intelligence.

## Results

Our sample included 1196 individuals (891 AD dementia patients and 305 cognitively unimpaired individuals) from four cohorts (Supplementary Table 1). The discovery and validation datasets consisted of 320 and 571 AD dementia patients, respectively. Cohort demographics are summarized in Table 1.

The longitudinal gray matter patterns that we estimated for the cognitively unimpaired (CU) and AD groups, show that the CU group deteriorates in gray matter with aging (Fig. 1A) and as expected that the AD group has more extensive atrophy (Fig. 1B). The correction method (gray matter of each AD patient standardized with respect to the CU model underlying Fig. 1A) that was applied to the AD dataset shows, at the population level, that AD presents with distinct atrophy patterns depending on the patient's age. Patients under 65 years of age typically have more posterior cortical atrophy, while patients over 75 years old show a prototypical AD mediotemporal atrophy pattern (Fig. 1C).

## Clustering evaluation

Longitudinal clustering showed that the 2-cluster and 5-cluster models were the most optimal with marginal differences. The 2-cluster model was preferable for one clustering criterion (fewer random effect parameters with high autocorrelation in their MCMC samples) while the 5-cluster model was more favorable for another (lower model deviance) (see Supplementary Table 2). The other clustering solutions had worse quality score combinations (either many autocorrelated MCMC samples or high model deviance) (Supplementary Table 2)[15]. The 2-cluster solution (Supplementary Fig. 1, fitted values) separated the discovery set only in terms of cortical severity (high versus low brain atrophy), whereas the 5-cluster solution (Fig. 2, fitted values) revealed spatially different atrophy subtypes. Since different spatial atrophy subtypes are of greater importance from an exploratory perspective and given the previous literature in AD subtypes[7], we chose to interpret the results of the 5-cluster solution.

## Cluster atrophy patterns and discriminant features

In the discovery dataset, we found five clusters of patients that showed gradual or steep longitudinal atrophy progression (Fig. 2). The largest cluster, minimal atrophy (MA, 59.1%), had very little mediotemporal atrophy at the clinical AD onset compared with the CU group (Fig. 2, 1.6 standard deviations below the CU population, Supplementary Fig. 2, 0.5 standard deviations below the CU population). It progressed slowly with entorhinal and hippocampal involvement that extended to other temporal lobe regions. The second largest cluster, limbic predominant atrophy (LPA, 29.1%), presented with entorhinal cortex atrophy at the clinical onset, with later involvement of other temporal lobe regions including the hippocampus. The third cluster, LPA+ (7.2%), was spatially similar to the LPA cluster but exhibited more atrophy in the entorhinal cortex at the AD onset. Atrophy progressively extended to the temporal lobe and then further to the rest of the cortex. We also found a cluster, diffuse atrophy (DA, 1.6%), with temporal and frontal involvement already at AD onset, where the atrophy diffused fast during the disease course. The last cluster, hippocampal sparing (HS, 3.1%), had parietal lobe atrophy and no involvement of medial-temporal lobe structures at disease onset, but fast atrophy progression. The MA and LPA patterns converged to widespread temporal lobe atrophy while the LPA+ converged to DA seven years after the disease onset. The most atypical atrophy pattern, HS, also progressed to a more diffuse atrophy pattern over time but with less involvement of the hippocampus. The cluster names were decided based on the atrophy pattern at AD onset. Table 2 provides a four-dimensional characterization of each subtype to illustrate how the patterns of atrophy and cognition evolve over time (see also Supplementary movie 1, Table 3, Fig. 3).

The cluster intercepts (AD onset) showed that the HS and DA clusters exhibit considerably thinner cortex in the parietal lobe than the other three clusters (Figs. 2 and 4). The LPA cluster has less entorhinal atrophy than the LPA+. Regarding the cluster slopes (atrophy evolution over time), the posterior cingulate gyrus, pars opercularis, pars-orbitalis gyri, and insula discriminate both DA and HS from the other three clusters (Figs. 2, 4, Supplementary Fig. 2). The atrophy slopes of the HS cluster were the steepest, followed by the DA and the LPA+ clusters.

The five longitudinal patterns of atrophy (Fig. 2) revealed a fine grouping that included variations in the stereotypical distribution of atrophy staging in AD[5] compared to the 2-cluster solution (Supplementary Fig. 1). In Table 3, we have summarized the longitudinal patterns of atrophy, to show the different features of the five longitudinal patterns and the patient characteristics related to them. After the main cluster analysis, the post hoc hierarchical clustering of cluster-specific atrophy intercepts and slopes (Fig. 4, slope dendrogram and figure legend) revealed quantitatively, that MA, LPA, and LPA+ have similar spatial distribution of atrophy over time (however, different atrophy

**Table 1 | Demographic and clinical characteristics of participants in the cohorts included in the training and validation cohorts**

| | Cognitively unimpaired | | | Alzheimer's disease patients | | | |
|---|---|---|---|---|---|---|---|
| | ADNI | J-ADNI | AIBL | ADNI | J-ADNI | AIBL | AddNeuroMed |
| **Discovery dataset** | | | | | | | |
| N[1] | 158 (52.1%) | 62 (20.2%) | 85 (27.7%) | 207 (64.7%) | 90 (28.1%) | 23 (7.2%) | – |
| Females[1] | 81 (51.3%) | 28 (45.2%) | 45 (52.9%) | 91 (44%) | 52 (57.8%) | 13 (56.5%) | – |
| Age at first visit[2] | 73.5 (5.7) | 67.5 (5.8) | 70.2 (7.4) | 75.7 (7.1) | 75 (8.6) | 72.6 (9.9) | – |
| AD clinical onset age, median[2] | – | – | – | 72 (7.4) | 73 (7.4) | 72 (7) | – |
| Education class[3] | 3.65 (0.66) | 3.15 (0.92) | 2.91 (1.02) | 3.36 (0.83) | 2.57 (0.91) | 2.61 (0.99) | – |
| APOE e4 allele carrier[1,4] | 26 (16.2%) | 8 (12.9%) | 26 (30.6%) | 155 (74.9%) | 55 (61.1%) | 17 (73.9%) | – |
| APOE e2 allele carrier[1,4] | 30 (18.8%) | 5 (8.1%) | 17 (20%) | 8 (3.9%) | 4 (4.4%) | 0 (0%) | – |
| MMSE total[2] | 30 (0) | 30 (0) | 29 (1.48) | 23 (2.97) | 23 (1.48) | 24 (4.45) | – |
| CDR[3] | 0 (0) | 0.01 (0.06) | 0.03 (0.12) | 0.79 (0.27) | 0.64 (0.23) | 0.67 (0.24) | – |
| CDR SOB[3] | 0.04 (0.13) | 0.05 (0.15) | 0.03 (0.12) | 4.54 (1.56) | 3.61 (1.40) | 3.85 (1.39) | – |
| **Validation dataset** | | | | | | | |
| N[1] | – | – | – | 216 (37.8%) | 168 (29.4%) | 67 (11.7%) | 120 (21%) |
| Females[1] | – | – | – | 97 (44.9%) | 98 (58.3%) | 38 (56.7%) | 79 (65.8%) |
| Age at first visit[2] | – | – | – | 76.4 (7.6) | 76.2 (5.6) | 76.7 (7.1) | 76 (6.6) |
| AD clinical onset age[2] | – | – | – | 74 (8.9) | 75 (5.9) | 74.9 (7.4) | 72 (5.9) |
| Education class[3] | – | – | – | 3.26 (0.91) | 2.57 (0.91) | 2.5 (1.06) | 1.54 (0.82) |
| APOE e4 allele carrier[1,4] | – | – | – | 115 (53.2%) | 74 (44%) | 24 (35.8%) | 59 (49.2%) |
| APOE e2 allele carrier[1,4] | – | – | – | 85 (39.4%) | 59 (35.1%) | 23 (34.3%) | 48 (40%) |
| Abeta positive[1] | – | – | – | 99 (45.8%) | 16 (9.5%) | 22 (32.8%) | – |
| Abeta negative[1] | – | – | – | 27 (12.5%) | 14 (8.3%) | 2 (3%) | – |
| MMSE total[2] | – | – | – | 23 (2.97) | 22 (2.9652) | 22 (4.44) | 22 (5.93) |
| CDR[3] | – | – | – | 0.81 (0.35) | 0.71 (0.25) | 0.84 (0.46) | 1.18 (0.50) |
| CDR SOB[3] | – | – | – | 4.63 (2.23) | 4.02 (1.42) | 4.93 (2.66) | – |

Notes: [1]n (%); [2]median (median absolute distance); [3]mean (sd); [4]the percentage denominator refers to the sum of the non-missing APOE records; education years are categorized in 4 classes (1 =< 0–8 years; 2 = 9–13 years; 3 = 13–15 years, 4 > 15 years). The MMSE scores at the baseline MRI visit for the AD group of the discovery set had a median value of 23 (1st quartile: 21, 3rd quartile: 25).
*MMSE* Mini-Mental State Examination, *CDR* Clinical Dementia Rating, *CDR SOB* CDR sum of boxes.

levels at the AD onset and different rates of atrophy progression) starting in the mediotemporal lobe and spreading further into the neocortex. The HS pattern follows another spatial atrophy distribution, starting in cortical regions. The DA cluster is quantitatively grouped together with the HS pattern but expresses both progression atrophy patterns since we observed it in a later disease stage (already widespread atrophy).

## Cluster characteristics
The percentages of patients from each cohort in the five clusters were similar (Table 3). In the discovery dataset, MA had the highest prevalence of *APOE* e4 carriers (75%), while HS had the lowest (40%). Patients in the DA and HS clusters had higher education levels (>15 years) followed by patients in the MA, LPA, and LPA+ (≤15 years). Using the MA (the largest cluster in the dataset) as reference group we found significantly lower American National Adult Reading Test (ANART) scores in LPA+ and HS ($p < 0.05$). Mini-Mental State Examination (MMSE) at AD onset was significantly worse for LPA ($p < 0.05$) (Fig. 3). Longitudinally, LPA+ and HS had the fastest decline in MMSE ($p < 0.05$). Regarding the Alzheimer's disease assessment scale (ADAS-cog) subscales, memory (word recall) was initially lower in LPA and LPA+ had the fastest decline over time in that domain. Language (following commands) and praxis (constructional) were significantly worse for the HS than the other clusters at AD onset. Orientation (ADAS) was worse for the LPA+ at AD onset.

In the model validation, no differences in amyloid-β (Aβ) status between clusters were found. Information regarding patient medical history was available for the Alzheimer's Disease Neuroimaging Initiative (ADNI) and the Japanese Alzheimer's Disease Neuroimaging

Initiative (J-ADNI), but not for the Australian Imaging, Biomarkers and Lifestyle study (AIBL) or the AddNeuroMed cohorts. A summary of the cluster medical history characteristics can be found in Supplementary Table 3. The distribution of disease duration at MRI visit for each cluster is presented in Supplementary Table 4.

## Intercept and slope covariance matrices
MA had the greatest total nodal strength and was used as a reference group for pairwise cluster comparisons of intercepts and slopes. The nodal strength of the LPA and LPA+ was lower with few exemptions (Fig. 5). The DA had higher nodal strength in only a few medial (frontal, temporal, and occipital) brain regions (intercepts and slopes) and the HS had higher nodal strength at the intercept of some ventromedial prefrontal and medial temporal regions. Cluster-specific intercept and slope covariance matrices are shown in Supplementary Fig. 3.

## Model validation
Our model was validated in two ways. First, we used an independent external dataset of unseen patient MRIs, to assess whether the classification of new data in one of the five longitudinal atrophy patterns yield sensible results. In addition to that we applied clustering separately to ADNI and J-ADNI/AIBL datasets.

The cluster probabilities show that few patients had a high probability of belonging to more than one clusters in the discovery dataset (Supplementary Table 5), and even fewer patients in the validation dataset (0.009% of the dataset, Supplementary Table 6, Supplementary Fig. 4). Finally, median cortical and hippocampal atrophy at the median disease duration for each cluster in the validation

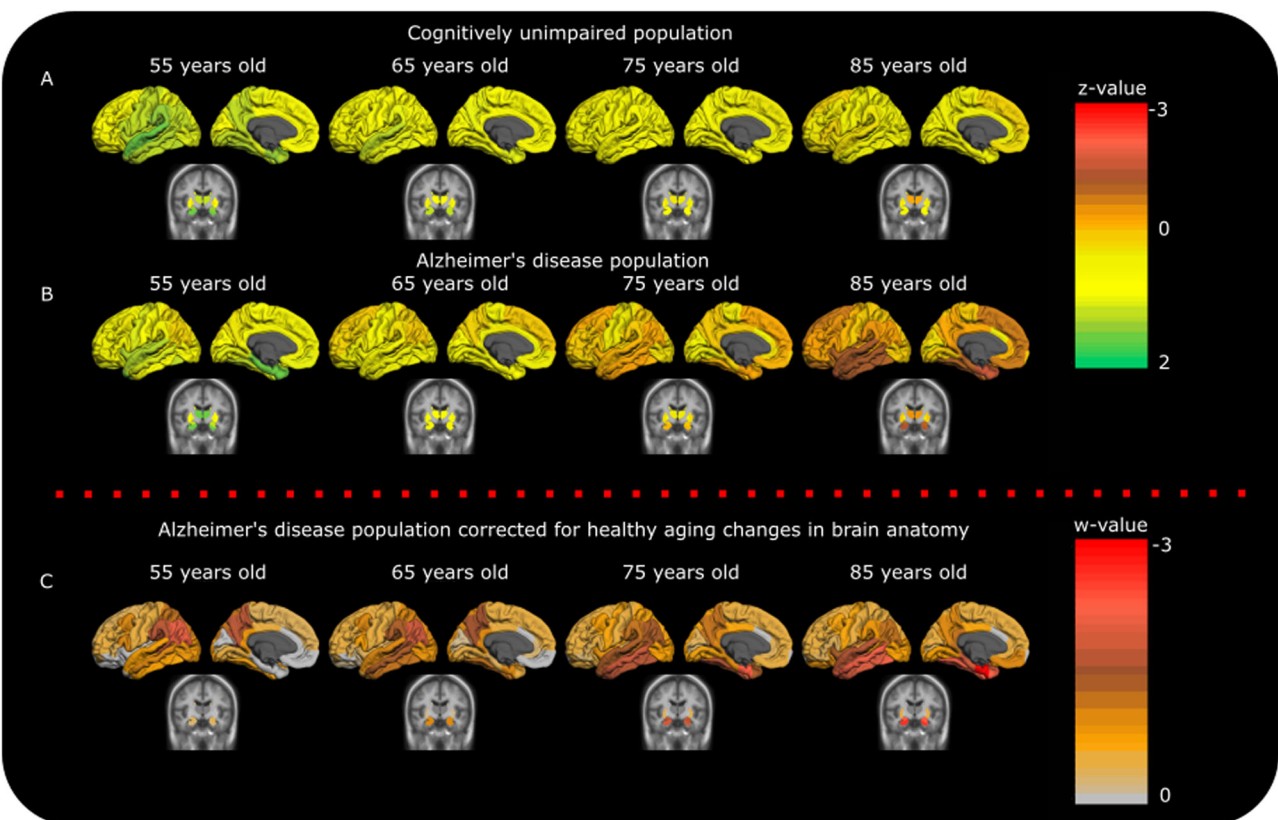

**Fig. 1 | Atrophy at population level in the CU and AD groups.** For the calculation of cognitively unimpaired (CU) and Alzheimer's disease (AD) atrophy patterns at the different ages (**A**, **B**), the data were z value transformed. One mixed effect multivariate model was used to visualize the differences in atrophy between the two diagnostic labels (red color; more atrophy, yellow color; less atrophy). The upper right color legend refers to standard deviations from the sample mean (0 corresponds to the mean of AD and CU sample values). At 55 years of age, AD has seemingly similar atrophy levels to the CU population and differences show up with ageing. For the visualization of the AD data correction based on the CU sample (**C**), two separate mixed effect multivariate models (one for the CU sample and one for the AD sample) were used. The AD data were standardized based on the CU data. Thus, the lower color legend shows standard deviations of the AD population below the CU population (w-values, 0 corresponds to the mean of CU sample values). Younger AD patients (between 55 and 65 years of age) show more posterior atrophy compared to controls, while older AD patients (above 75 years old) show more mediotemporal and hippocampal atrophy compared to controls. For the visualization models the fitted values in panels **A**, **B**, and **C** are controlled for MRI field strength, cohort.

dataset showed high similarity to the model's fitted values at the same disease stage (Fig. 6, Supplementary Fig. 5).

Moreover, when clustering was applied separately to ADNI and J-ADNI/AIBL datasets, the former dataset showed five and the latter dataset showed four different patterns of atrophy (Supplementary Figs. 6 and 7 (uncorrected version of Supplementary Fig. 6), Supplementary Tables 7–8). The atrophy patterns found in the separate cohorts were similar to the overall discovery dataset, including MA, LPA, LPA+, DA, and HS cases. Quantitatively, MA is more similar (in terms of intercept and slopes distances) to the ADNI cluster 3 and J-ADNI/AIBL cluster 3, LPA is more similar to ADNI cluster 2 and J-ADNI/AIBL cluster 2, LPA+ is more similar to ADNI cluster 1 and J-ADNI/AIBL cluster 3, DA is more similar to ADNI clusters 4 and J-ADNI/AIBL cluster 4, and finally HS is more similar to ADNI cluster 2 and J-ADNI/AIBL cluster 1 (Supplementary Figs. 6–7, Supplementary Tables 7–8). The similarities between the ADNI and J-ADNI/AIBL datasets can be found in the supplementary analysis (Supplementary Fig. 7).

## Discussion

A major contribution of this study is the transition from a cross-sectional understanding of AD subtypes to the perspective brought by longitudinal clustering. Some of the previously reported AD subtypes seem to reflect different stages of the disease that can be observed in our five estimated longitudinal atrophy patterns. Hence, our data contribute a step towards solving the long-lasting problem of

disentangling disease stages from actual disease subtypes. This was enabled by modeling longitudinal data using a clear timescale, i.e., over eight years, from disease onset in a large multiethnic cohort of 891 AD dementia cases from four continents. Another important finding is that AD subtypes with clearly distinct atrophy trajectories may converge in late disease stages. This introduces a new understanding of neurodegeneration in AD, which combined with knowledge of neuropathological and clinical heterogeneity, could set the ground for future personalized predictions of biological changes and cognitive decline in AD.

At the modeled clinical disease onset, our method successfully identified the same patterns of atrophy previously identified in neuropathological and neuroimaging subtyping studies (minimal atrophy, limbic predominant, typical AD, and hippocampal sparing)[5,7,8,13,16]. Our results revealed two main pathways of atrophy. We introduce the term pathway to describe AD patients that show similar spatial distribution of atrophied brain regions over time. Within the same atrophy pathway, patients may progress faster (LPA+) than others (LPA and MA) but their spatial distribution of atrophy over time is similar. This pathway contrasts with the second different atrophy pathway in AD, which has a different spatial distribution with mainly cortical atrophy over time. The differences in progression rates also reflect the rates of cognitive decline of the patients. It is a very important future aim to understand the factors underlying of these differences in progression within the same pathway but also between the different pathways that we have identified.

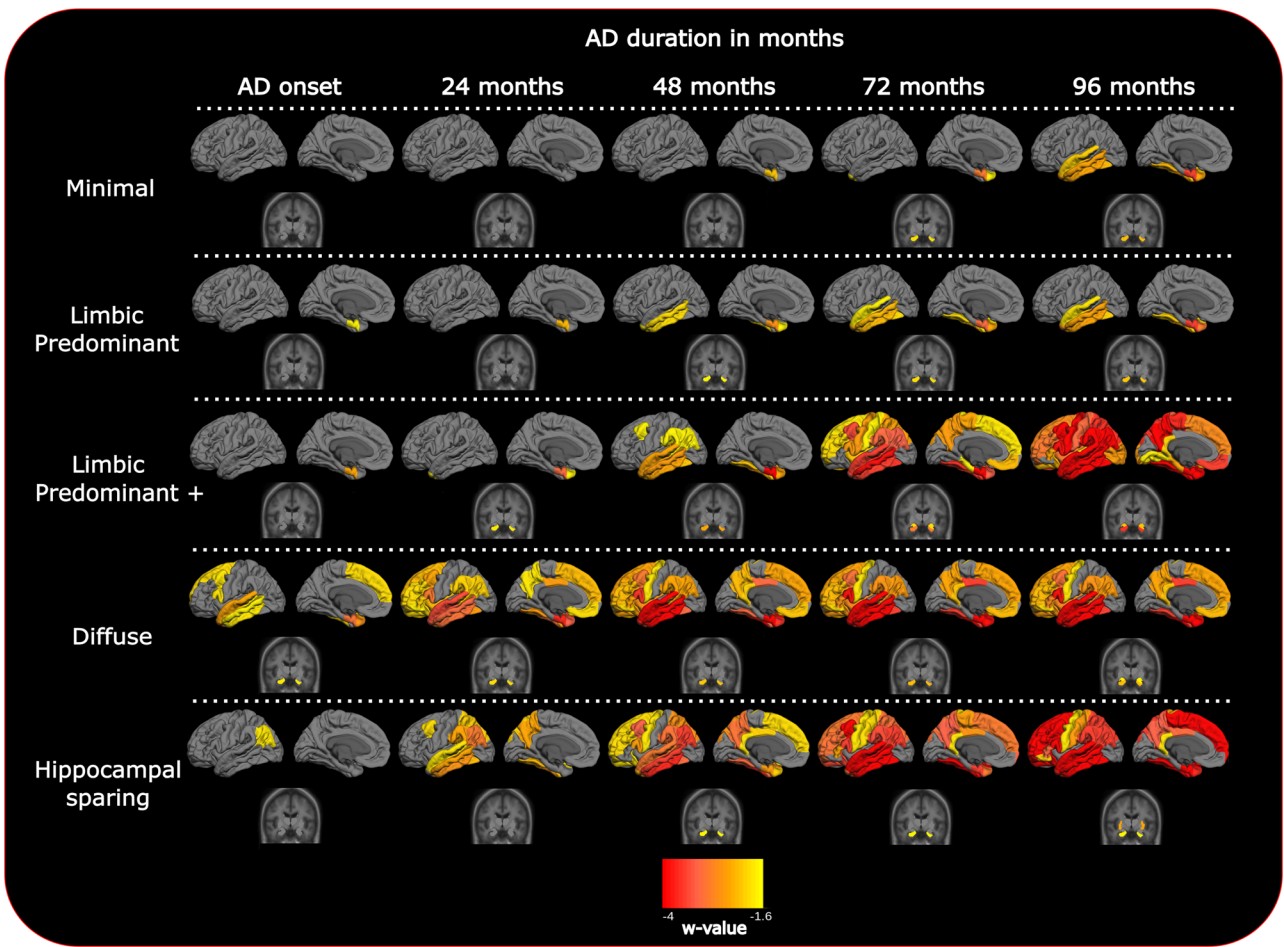

**Fig. 2 | Fitted values for cortical thickness and subcortical volumes for the different longitudinal patterns of atrophy from AD onset.** Atrophy-fitted values from clinical AD onset. Each row represents one cluster of patients with the corresponding pattern of atrophy. The color scale illustrates cortical thinning and subcortical volume loss compared to Aβ negative, cognitively unimpaired (CU) individuals (red color; more atrophy, yellow color; less atrophy). Data are w-value transformed and therefore colors represent standard deviations below the CU group controlled for aging. Fitted values are fixed for intracranial volume and MRI scanner field strength.

## Table 2 | Cluster/cognitive profiles summary

| Patterns of atrophy acronym | Atrophy at the AD onset | Atrophy rate | Cognitive decline rate | Atrophy 8 years after the AD onset |
|---|---|---|---|---|
| MA | Minimal | Slow | Slow | Limbic predominant |
| LPA | Limbic predominant | Slow | Slow | Limbic predominant |
| LPA+ | Limbic predominant | Fast | Fast | Diffuse |
| DA | Diffuse | Fast | Slow | Diffuse |
| HS | Hippocampal sparing | Fast | Fast | Cortical predominant |

Summary of longitudinal atrophy and cognitive trajectories of AD patients in four cohorts (Alzheimer's Disease Neuroimaging Initiative, Japanese ADNI, AddNeuroMed, and the Australian Imaging, Biomarkers, and Lifestyle study).

The minimal atrophy (atrophy limited to the entorhinal cortex), the limbic predominant (atrophy mainly in limbic areas), and the typical (widespread atrophy in the hippocampus, temporal, parietal, and frontal lobes) AD subtypes[16], were identified in some disease stage of our MA, LPA, or LPA+ longitudinal atrophy clusters. MA was the most representative cluster in the datasets under investigation and it had the highest variability within cluster. Clustering methods often identify one cluster that represents the most prevalent pattern in a dataset which is an average of more heterogeneous observations than the pattern that results from the remaining clusters in the dataset[16]. It is important to stress that our MA cluster includes patients that are grouped in the minimal and limbic predominant patterns of atrophy, and potentially some early stage typical AD

patients reported in the literature[7]. This is the case, since in our study we model trajectories of atrophy from the disease onset accounting for longitudinal structural changes in CU *Aβ* negative subjects. Through this type of modeling, we connected patterns of atrophy from the literature by modeling atrophy trajectories and therefore disease staging explicitly. Our MA and LPA clusters probably belong to the same AD subtype observed in two distinct stages, since MA patients reached the LPA levels (baseline) two years after the AD onset. The differences in cognitive intercepts (MMSE and ADAS word recall) between our MA and LPA clusters support the view that they reflect different disease stages. The LPA+ cluster appears to be on the same atrophy pathway but with faster atrophy rates in comparison to the MA and LPA clusters. Patients in the LPA+ cluster had the steepest

**Table 3 | Cluster characteristics**

| | Minimal | | Limbic predominant | | Limbic predominant + | | Diffuse | | Hippocampal sparing | |
|---|---|---|---|---|---|---|---|---|---|---|
| | Discovery | Validation | Discovery | Validation | Discovery | Validation | Discovery | Validation | Discovery | Validation |
| N[1] | 189 (59.1%) | 313 (54.8%) | 93 (29.1%) | 230 (40.3%) | 23 (7.2%) | 11 (1.9%) | 5 (1.6%) | 11 (1.9%) | 10 (3.1%) | 6 (1.1%) |
| Females[1] | 90 (47.6%) | 173 (55.3%) | 45 (48.4%) | 126 (54.8%) | 14 (60.9%) | 5 (45.5%) | 3 (60%) | 4 (36.4%) | 4 (40%) | 4 (66.7%) |
| Cohort ADNI[1] | 119 (63%) | 118 (37.7%) | 61 (65.6%) | 87 (37.8%) | 17 (73.9%) | 5 (45.5%) | 4 (80%) | 4 (36.4%) | 6 (60%) | 2 (33.3%) |
| Cohort J-ADNI[1] | 60 (31.7%) | 90 (28.8%) | 21 (22.6%) | 69 (30%) | 6 (26.1%) | 5 (45.5%) | 1 (20%) | 3 (27.3%) | 2 (20%) | 1 (16.7%) |
| Cohort AIBL[1] | 10 (5.3%) | 33 (10.5%) | 11 (11.8%) | 30 (13%) | – | – | – | 2 (18.15%) | 2 (20%) | 2 (33.3%) |
| Cohort AddNeuroMed[1] | – | 72 (23%) | – | 44 (19.2%) | – | 1 (9%) | – | 2 (18.15%) | – | 1 (16.7%) |
| Age[2] | 75.5 (7.6) | 76.5 (6.7) | 75.4 (8) | 76.3 (6.7) | 74.5 (7.3) | 75.1 (7.9) | 71.9 (11.6) | 73.3 (5) | 62.7 (7.9) | 62.6 (7.3) |
| AD onset age[2] | 73 (7.4) | 74 (7.4) | 73.1 (7.6) | 75 (7.3) | 73 (7.4) | 72 (7.4) | 66 (7.4) | 73 (4.4) | 61 (8.1) | 60.9 (6.5) |
| Education class[3] | 3.08 (0.93) | 2.6 (1.13) | 3.04 (1) | 2.59 (1.08) | 3 (0.9) | 2.73 (1.01) | 3.6 (0.55) | 2.82 (1.08) | 3.5 (0.85) | 3 (1.26) |
| APOE e2[1,5] | 6 (3%) | 17 (6%) | 3 (3%) | 13 (6%) | 1 (4%) | 1 (10%) | 1 (20%) | 1 (9%) | 1 (10%) | 1 (20%) |
| APOE e3e3[1,5] | 45 (24%) | 99 (33%) | 29 (31%) | 77 (34%) | 7 (30%) | 1 (10%) | 1 (20%) | 5 (45.5%) | 5 (50%) | 0 |
| APOE e4[1,5] | 142 (75%) | 183 (60%) | 62 (67%) | 139 (62%) | 16 (70%) | 8 (80%) | 3 (60%) | 5 (45.5%) | 4 (40%) | 4 (80%) |
| ANART[4] | 16.74 (0.42) | | 16.13 (0.58) | | 12.08[a] (1.07) | | 16.85 (2.2) | | 12.57[a] (1.8) | |
| MMSE total[5] | 23.11[c] (−0.42) | | 22.51[a] (−0.5) | | 24.95 (−1.99[b]) | | 21.83 (−0.5) | | 21.78 (−1.59[b]) | |
| CDR[3] | 0.72 (0.27) | 0.86 (0.40) | 0.75 (0.25) | 0.89 (0.45) | 0.76 (0.26) | 0.75 (0.27) | 0.67 (0.29) | 1 (0.61) | 0.71 (0.27) | 0.88 (0.25) |
| CDR SOB[3] | 4.02 (1.55) | 4.34 (2.11) | 4.28 (1.48) | 4.40 (1.95) | 4.45 (1.73) | 4.64 (0.80) | 3.83 (3.01) | 4.93 (2.22) | 4.29 (0.99) | 4.67 (1.15) |
| GDS[5] | 2.06 (0.02) | | 2.13 (−0.01) | | 2.19 (0.09) | | 0.84 (0.11) | | 1.96 (−0−01) | |
| **Alzheimer's disease assessment scale** | | | | | | | | | | |
| Word recall[5] | 5.96 (0.09) | | 6.35[a] (0.04) | | 5.5 (0.55[b]) | | 7.01 (−0.03) | | 6.43 (0.36) | |
| Following commands[5] | 0.61 (0.01) | | 0.77 (0) | | 0.5 (0.11) | | 1.39 (−0.1) | | 1.7[a] (0.04) | |
| Constructional praxis[5] | 0.71 (0.01) | | 0.92 (−0.01) | | 0.7 (0.1) | | 0.89 (−0.08) | | 1.9[a] (0.01) | |
| Naming objects and fingers[5] | 0.39 (0.01) | | 0.51 (0.01) | | 0.28 (0.14) | | 0.51 (0) | | 1.13 (−0.23) | |
| Ideational praxis[5] | 0.49 (0.01) | | 0.61 (0.01) | | 0.41 (0.07) | | 0.6 (−0.05) | | 1.5 (0.01) | |
| Orientation[5] | 2.24 (0.17) | | 2.71 (0.08) | | 2.02 (0.75[b]) | | 2.39 (0.17) | | 2.4 (0.35) | |
| Word recognition[5] | 5.96 (0.13) | | 5.89 (0.1) | | 4.92 (0.5) | | 6.08 (0.01) | | 3.16 (0.98) | |
| Recall instructions[5] | 0.27[c] (0.05) | | 0.51 (0.03) | | 0.1 (0.06) | | 0.27 (0.04) | | 0.23 (0.19) | |
| Spoken language[5] | 0.31 (0.02) | | 0.44 (0.03) | | 0.15 (0.12) | | 0.52 (−0.02) | | 0.8 (−0.02) | |
| Word finding difficulty[5] | 0.63 (0.03) | | 0.87 (0.02) | | 0.29 (0.2) | | 0.43 (0.05) | | 1.61 (−0.17) | |
| Comprehension of spoken language[5] | 0.3 (0.02) | | 0.49 (0.01) | | 0.22 (0.09) | | 0.33 (−0.02) | | 0.54 (0) | |

Notes: [1]n (%); [2]median (median absolute distance); [3]mean (sd); [4]estimated value at the AD onset (estimation standard error); [5]estimated value at the AD onset (estimated annual change); [5]the percentage denominator refers to the sum of the non-missing APOE records; education years are categorized in 4 classes (1 = <0–8 years; 2 = 9–13 years; 3 = 13–15 years, 4 > 15 years). Corrections for multiple comparisons were assessed with the Holms–Sidak method.
*ANART* American National Adult Reading Test, *MMSE* Mini-Mental State Examination, *CDR* Clinical Dementia Rating, *CDR SOB* CDR sum of boxes, *GDS* Geriatric Depression Scale.
[a]Baseline differences between cluster 1 and other clusters.
[b]Longitudinal differences between cluster 1 and other clusters.
[c]Baseline or longitudinal differences between discovery and validation dataset.

decline in cognition among the five identified clusters, including memory and orientation. LPA+ patients had similar *APOE e4*[7], education and disease onset as in MA and LPA. However, premorbid intelligence, a proxy for cognitive reserve[17], was significantly higher in LPA+ than in MA and LPA. We believe that due to high cognitive reserve, patients of the LPA+ cluster can reach higher levels of brain atrophy than the MA and LPA clusters, while maintaining similar clinical severity until they reach the AD onset[17]. The dynamics of brain atrophy over time in the MA, LPA, and LPA+ clusters differed. However, our current data seems to indicate that these three longitudinal atrophy clusters belong to the same atrophy pathway in AD, namely the mediotemporal atrophy pathway. Atrophy in this well-documented pathway is shown to correlate with the

neurofibrillary tangle pathology at autopsy[1,5,18]. Even though these three clusters (MA, LPA, and LPA+) belong to the same atrophy pathway, their rates of atrophy and cognitive decline differ substantially, which can have important clinical implications. These observed differences are likely due to a combination of protective and risk factors as well as potential concomitant non-AD brain pathologies[7]. For example, it was shown by Ferreira and colleagues, that the location and frequency of markers of small vessel disease differ between AD subtypes[19].

Our HS cluster resembles the hippocampal sparing subtype described in previous neuropathological and neuroimaging subtyping studies[5,7,8,13,16]. This subtype is more often characterized by cortical atrophy in comparison to the other AD subtypes[7,8,16,18]. In our study,

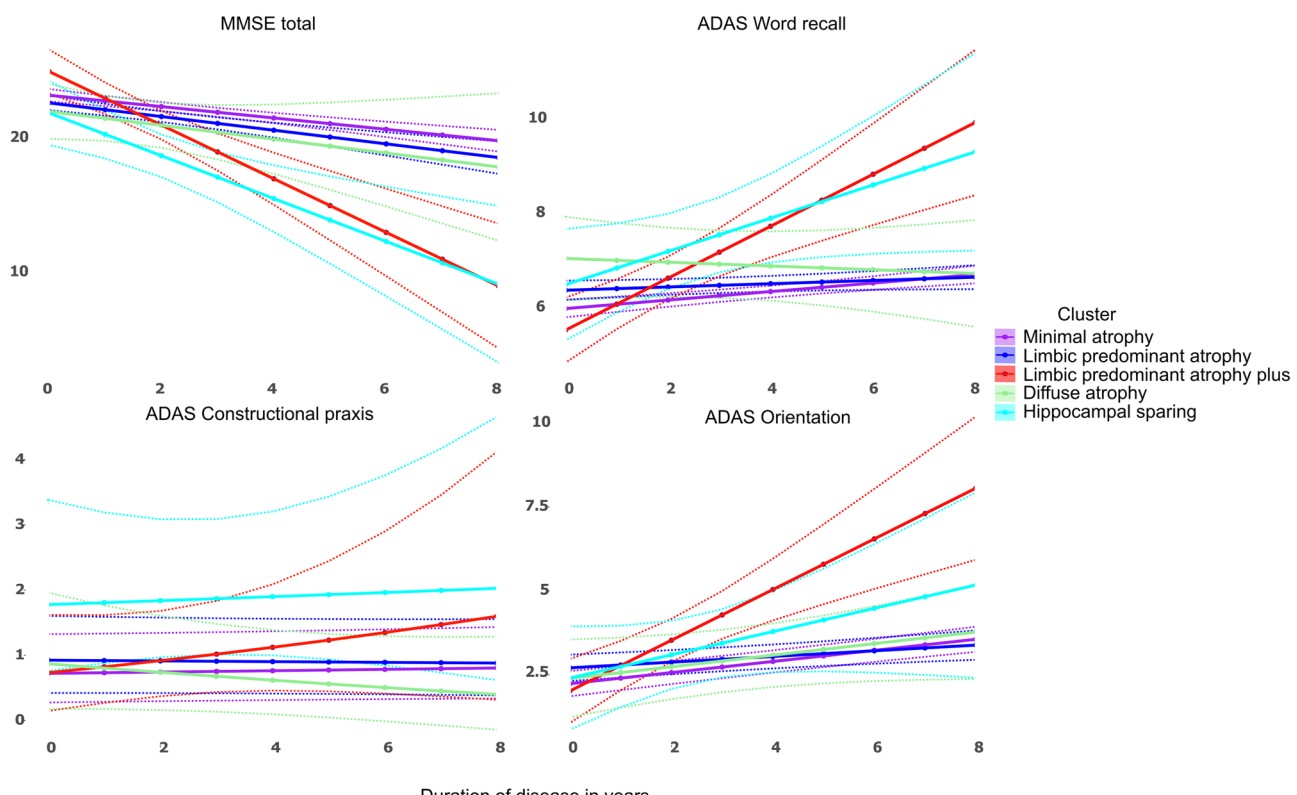

**Fig. 3 | Cluster-specific cognitive trajectories after the clinical onset of dementia.** The trajectories are estimated with mixed effect models to account for intra subject and cohort variability. MMSE Mini-Mental State Examination, ADAS Alzheimer's disease assessment scale. Dotted lines represent 95% confidence intervals.

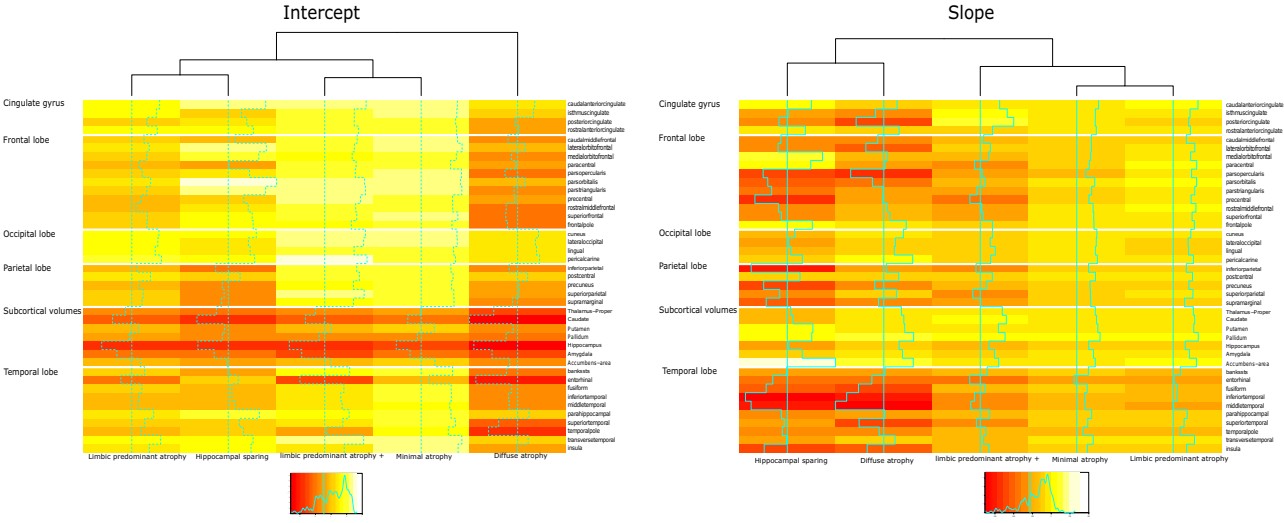

**Fig. 4 | Longitudinal clustering model cluster-mean intercept and slope atrophy coefficients.** Each row of the heatmap is grouped in terms of neuroanatomical spatial position (red color; more atrophy, yellow color; less atrophy). Columns that represent different clusters are grouped in terms of similarity between clusters. Vertical lines within cells represent cluster region mean ROI value (the vertical dotted line represent the value 0, no difference from the CU sample). The diffuse atrophy cluster has the lowest intercept and it is not grouped with any other cluster. The cluster slopes of the diffuse atrophy and hippocampal sparing clusters are grouped together. The minimal atrophy and limbic predominant atrophy/Limbic predominant plus are grouped together.

some characteristics of the HS cluster included steep atrophy trajectories, a lower frequency of the *APOE* e4 allele[7], high premorbid intelligence, more years of education, and early AD onset, which is in line with the characteristics associated with the hippocampal sparing subtype reported by previous studies[7,8,16]. This cluster had the lowest frequency, which is also in line with previous studies[7,8]. The chances of finding more hippocampal sparing patients were reduced since the cohort selection criteria included the amnestic phenotypic presentation of AD, which is frequently related to typical AD and thus the mediotemporal atrophy pathway[4]. The significantly affected constructional and ideational praxis is a key characteristic of the hippocampal sparing subtype[7,13,16], which was also confirmed in our study. Comparisons between our MA and HS cluster covariance patterns revealed network differences between these two groups. In the MA,

## Minimal vs Limbic predominant

A) Intercept

B) Slope

## Minimal vs Limbic predominant +

C) Intercept

D) Slope

## Minimal vs Diffuse

E) Intercept

F) Slope

## Minimal vs Hippocampal sparing

G) Intercept

H) Slope

*Higher in other Clusters*          *Higher in minimal atrophy*

**Fig. 5 | Comparison of cluster-specific covariance matrixes with node strength.** The cluster-specific intercept (**A**, **C**, **E**, and **G**) and slope (**B**, **D**, **F**, and **H**) covariance matrices were compared with network theory. Sphere diameter shows the node strength of each region. The regions where the minimal atrophy cluster has higher nodal strength than the other clusters are shown in red, while blue is used in the opposite case. The networks are presented in lateral sagittal and transversal view.

anatomical differences due to the disease were predominantly localized in the medial-temporal lobe and cortical regions combined as a network at the AD onset. On the other hand, the HS cluster network differences at the AD onset also involve the basal ganglia. Moreover, the HS cluster had higher nodal strength at the intercept of some ventromedial prefrontal and medial temporal regions from the MA

cluster. Based on all these results, we believe that the HS pattern of atrophy represents a distinct atrophy pathway in AD, namely the cortical pathway.

To explain the atrophy trajectories of our DA cluster is challenging since excessive frontal and temporal atrophy was already present at the clinical onset. Our data showed that in advanced stages on the

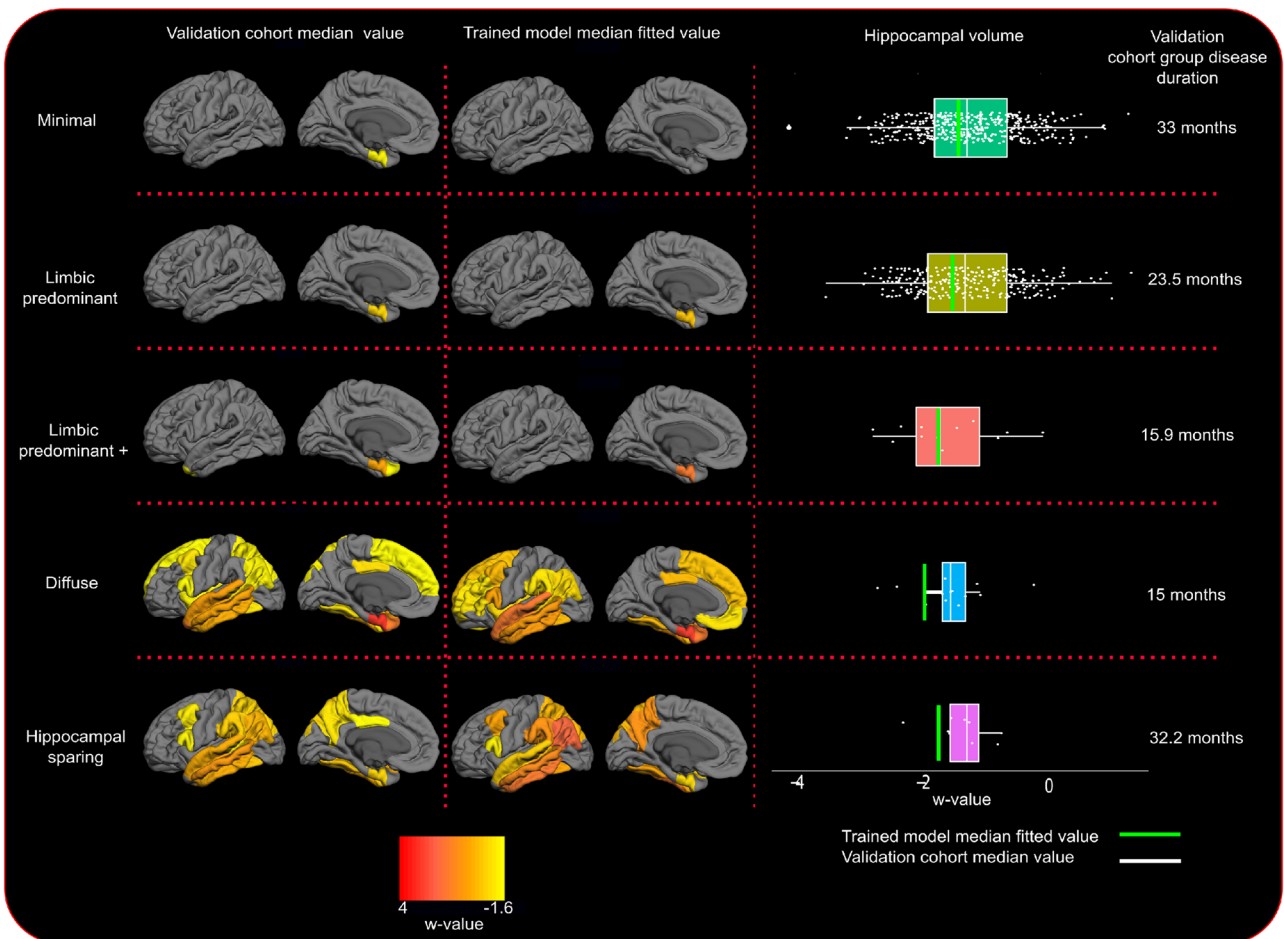

**Fig. 6 | Comparison of model-fitted values and validation dataset atrophy levels.** Atrophy-fitted values after the AD onset for the trained clustering model versus the new validation dataset. New observations were classified to each cluster, and median disease duration was calculated. Then atrophy-fitted values at the median disease duration of each cluster were calculated through the clustering model (middle column). Median atrophy maps (group median atrophy) for the new data of each cluster are presented in the left column. The right column shows the hippocampal volumes of each cluster's (boxplot colors: green; minimal; $n = 420$, olive; limbic predominant; $n = 283$, orange; limbic predominant+; $n = 12$, blue; diffuse; $n = 13$, purple; hippocampal sparing; $n = 8$) new observations (repeated measurements are included) and the model-fitted value hippocampal atrophy (green vertical line). The color scale of the cortical maps (left and middle column) reflects AD atrophy levels compared to a multicohort dataset of A$\beta$ negative cognitively unimpaired (CU) individuals (red color; more atrophy, yellow color; less atrophy). Data are w-value transformed and therefore colors represent standard deviations bellow the CU group controlled for aging. Fitted values are fixed for intracranial volume and MRI scanner field strength.

mediotemporal and cortical pathways of atrophy, AD patients may develop comparable levels of atrophy that are similar to our DA cluster. As a result, this cluster of patients can potentially belong to either of the two pathways of atrophy. Similarly to our LPA+, cognitive reserve in our DA cluster (education exceeded 15 years on average) may explain the greater atrophy levels (at dementia onset)[7,17]. Our DA cluster had a similar pattern of atrophy to that of the typical AD atrophy subtype reported in the literature[7,8,13,16], but lower frequency. In a recent cross-sectional clustering study using tau PET that mainly included preclinical AD, no cluster had spatial tau distribution similar to the typical AD pattern of atrophy, but the cortical and medial-temporal patterns of tau were observed[10]. Further, two other studies in prodromal AD found clusters of individuals with decreased temporal-parietal glucose metabolism[20] or increased temporal-parietal atrophy[21] (typical AD pattern), but in low sample frequencies, which is in line with our findings.

Recently, it was proposed that A$\beta$ aggregation in the default mode network (DMN) is predominantly associated with within-network but distant glucose hypometabolism[22]. Moreover, glucose metabolism, atrophy, and tau pathology are closely linked in AD[7,18,22]. We speculate that the mediotemporal path of neurodegeneration in AD may be

initiated in the vulnerable temporal lobe after enough A$\beta$ is deposited in distant DMN regions. In contrast, the cortical atrophy pathway patients may show less initial temporal lobe atrophy (and amnestic symptomatology) partially because they respond differently to A$\beta$ aggregation in the DMN due to compensation mechanisms[22] such as cognitive reserve[17].

Our study has addressed some important methodological challenges that the existing literature of biological subtypes has not overcome so far. To our knowledge, this is the first time that AD atrophy subtypes were discovered based on modeling longitudinal biomarker trajectories[8]. An immediate advantage of our longitudinal clustering approach is that it overcomes the assumption that subjects of a cluster (cross-sectional analysis) remain in the same cluster when the disease advances, which is unrealistic[8]. Previous studies have employed arbitrary timescales to model biomarker progression[8,10,13]. Our estimates are based on a clearly defined timescale, namely the time from clinical onset. This approach provides the unique possibility to generate interpretations based on disease staging that help to trace abnormal changes early in the disease course of each cluster. Previously, longitudinal interpretations could not directly relate back to data in hand

because they were not anchored to a specific timescale[13]. We calculated atrophy w-values for each patient corrected for the effects of aging in brain morphology based on a dataset of longitudinal $A\beta$ negative CU individuals. Our model for the correction of ageing effects on the atrophy values, as it was shown in the results, identified the excess atrophy due to AD at different ages correctly and is in line with the literature comparing early and late onset AD[23]. This approach helped to estimate the within-subject variance more precisely and therefore account for the effects observed in aging[9,15,24], which has been a limitation of cross-sectional estimations[9,16,18]. A common pitfall of clustering studies is to focus on finding labels for observations depending on their features in a population, which tends to overfit the training set. External validation datasets help to assess the ability of clustering models to generalize[8]. We found that our longitudinal atrophy estimates and the unseen atrophy patterns in the validation dataset were highly concordant. Moreover, the application of longitudinal clustering separately in the ADNI and J-ADNI/AIBL cohorts showed similar longitudinal atrophy patterns to those found in the whole discovery dataset with small variations. The low sample percentages that some clusters exhibited, is attributed to the under-representation of rare subtypes in some cohorts that focused on the typical AD phenotype, the lower sample that was used in the separate cohorts for clustering, and to the ability of our method to identify clusters of very low prevalence if they exist[15]. Concordance was high for the most prevalent atrophy patterns and lower for DA and HS, due to low sample sizes and cohort differences. Between ADNI and J-ADNI/AIBL cohorts, a quantitative assessment showed increased similarity in longitudinal atrophy trajectories, with small variations due to small sample sizes and cohort variability. Of interest, the hippocampal sparing and diffuse atrophy patterns of atrophy were found in both datasets but with lower prevalence than in the complete discovery dataset. This happened due to the split of the discovery dataset in smaller datasets that under-represent the AD population. AD subtypes of lower prevalence in the population[7], are doomed to be underrepresented or disappear when clustering is applied to small datasets[9]. The combined analysis of the cohorts in the discovery dataset with one model instead of building one clustering model per cohort, allowed us to build a single statistical model that produced more accurate estimates due to a larger sample size. Importantly, since our study was mainly based on longitudinal information from repeated cross-sectional measurements, we avoided to interpret structural relations between brain regions based on cross-sectional correlations. Instead, we focused only on the longitudinal correlation between brain regions which is based on within patient longitudinal trajectories.

Our study has some limitations. Only atrophy markers were modeled in the context of AD heterogeneity. Pre-AD scans were not included. This reduced our ability to infer atrophy patterns that precede the diagnosis of AD dementia. In the future, we envision combining and comparing other imaging modalities longitudinally, thus extending our current analyses to incorporate information about tau-related pathology. Moreover, the future addition of biomarkers of non-AD pathologies in the clustering studies design will help in understanding the contribution of comorbidities in AD subtypes. The inclusion of subjects from four different continents is a strength since it increased the variability in the sample and therefore represented the AD population better, but it is also a limitation due to variability in MRI assessments. Another limitation is the short follow-up period for AD patients included in the study. A future re-estimation of atrophy trajectories will include more MRI visits per patient to obtain better estimates. However, a strong methodological aspect of this study is the reconstruction of longitudinal subtype-atrophy profiles over the dementia part of the AD continuum, based on longitudinal individual patients' data that comprised short segments of the disease continuum. Future studies should also include multiple MRIs from patients that are followed up from the preclinical until the dementia stage. The cohorts were harmonized to reduce MRI variability. Beyond these limitations, we assumed that the CU population has homogeneous brain morphology. Future studies should investigate whether CU individuals age differently and incorporate this information in the context of AD heterogeneity.

In conclusion, based on a large multiethnic cohort of AD dementia patients, we discovered five longitudinal patterns of brain atrophy that group the previously reported AD subtypes into two atrophy pathways (a mediotemporal and a cortical). We introduced a different understanding of the neurodegenerative aspect of AD heterogeneity, by shifting from the cross-sectional understanding of AD subtypes to the perspective brought by longitudinal clustering. Our study is a step forward toward answering an urgent question, whether the observed heterogeneity in AD reflects disease stages or distinct biological subtypes. We believe that with the help of our proposed model, it will be possible to unravel the heterogeneity in AD, thus enabling precision medicine and potentially leading to successful disease-modifying treatments in the future.

## Methods
### Study design and participants
This study includes 891 AD dementia patients and 319 CU individuals from four international multi-center cohorts: Alzheimer's Disease Neuroimaging Initiative (ADNI, http://adni.loni.usc.edu) Japanese ADNI (J-ADNI, https://humandbs.biosciencedbc.jp/en/hum0043-v1)[25], AddNeuroMed (https://consortiapedia.fastercures.org/consortia/anm/)[26], and the Australian Imaging, Biomarkers and Lifestyle study (AIBL, Australian ADNI, https://aibl.csiro.au/)[27] (Table 4). The AD inclusion criteria of the four cohorts were similar since the research protocols of J-ADNI, AIBL, and AddNeuroMed were designed to be comparable with ADNI (Supplementary material, p. 1–2). All participants provided written informed consent in accordance with the Helsinki declaration and approval for the studies was obtained by the local ethics committees.

Only $A\beta$ positive AD patients (ADNI, J-ADNI, AIBL) were included in the discovery cohort to ensure that the identified clusters reflect AD pathology (Table 4). CU individuals were $A\beta$ negative (to exclude preclinical AD) and remained CU during all future cognitive assessments available to date (Table 4). Participants in the discovery dataset had more than two MRI visits (Supplementary Table 1, Supplementary Fig. 8), while those in the validation dataset had at least one visit (ADNI, J-ADNI, AIBL, AddNeuroMed). Some patients from the validation dataset (AddNeuroMed) had access to more than one MRI visit (Supplementary Table 1).

### Magnetic resonance imaging (MRI)
The J-ADNI, AddNeuroMed, and AIBL cohorts adopted the MRI protocol of ADNI. High resolution sagittal (1.5 T and 3 T) 3D T1-weighted Magnetization Prepared Rapid Gradient Echo (MPRAGE) volumes, with full brain and skull coverage were acquired and detailed quality control (QC) was applied to the original images. Images were processed with the longitudinal stream of FreeSurfer 6.0, through the TheHiveDB[28]. The parcellation and segmentation of MRIs with Freesurfer were QCed manually by a trained person to exclude bad segmentations/parcellations that would introduce noise to the results. Thickness from 34 cortical (Desikan atlas) and volumes of seven subcortical regions per hemisphere (Supplementary Table 9) were extracted and averaged between hemispheres. These regions were used as input for clustering. Estimated total intracranial volume (eTIV) was also extracted to account for differences in head size in volumetric measures.

**Table 4 | Characteristics of the cohorts included in the study**

| | ADNI | J-ADNI | AIBL | AddNeuroMed |
|---|---|---|---|---|
| Study design | Longitudinal multi-center cohort | Longitudinal multi-center cohort | Longitudinal multi-center cohort | Longitudinal multi-center cohort |
| Data collection period | 2004-present | 2018-2014 | 2006-present | 2004-2010 |
| Country | USA and Canada | Japan | Australia | Finland, France, Greece, Italy, Poland, Sweden, and United Kingdom |
| Inclusion criteria | MMSE (20-26); CDR (0.5-1); age (>65); NINCDS/NINCDS/ADRDA criteria for probable AD; GDS < 6 | MMSE (20-26); CDR (0.5-1); age (>65); NINCDS/ADRDA criteria for probable AD; GDS < 6 | MMSE (18-26); CDR (>0.5); NINCDS-ADRDA for criteria for probable AD; ICD-10 | MMSE (12-28); age (>60); NINCDS/ADRDA for criteria for probable AD and DSM-IV |
| Follow-up intervals | 3-12 months | 6-12 months | 18 months | 12 months |
| Structural MRI field strength | 1.5 T and 3 T | 1.5 T | 1.5 and 3 T | 1.5 T |
| Biomarkers available | Yes | Yes | Yes | No |
| CSF platform cut-off | Elecsys, <880 pg/mL[49a] | Luminex, <333 pg/mL[25] | – | – |
| PET cut off | FBB SUVR >1.08 AV45 SUVR >1.11[b] | AV45 SUVR >1.11[25,50], PiB SUVR >1.48[25,51] | PiB SUVR >1.5[52] | – |

*MMSE Mini-Mental State Examination, CDR Clinical Dementia Rating, GDS Geriatric Depression Scale, NINCDS/ADRDA National Institute of Neurological and Communicative Disorders and Stroke–Alzheimer's Disease and Related Disorders Association, ICD-10 Classification of Mental and Behavioral Disorders, SUVR standardized uptake value ratio, AV45 florbetapir, PiB 11C-Pittsburgh compound B, FBB Florbetaben.*
*[a]The UPENN biomarker batch 9 was used to quantify ADNI CSF positivity.*
*[b]Composite SUVRs for FBP ≥1.11 or FBB ≥1.08 were defined as positive as described on the ADNI website (https://adni.bitbucket.io/reference/docs/UCBERKELEYFBB/UCBerkeley_FBB_Methods_04.11.19.pdf).*

## Longitudinal clustering analysis

Statistical analysis consisted of three steps (Supplementary Fig. 9). In the first step, we estimated mean volume/thickness levels of the CU individuals (for the age span 50–90) in the discovery dataset based on linear mixed effect models. This was followed by calculations of w-values[29], which are z-values adjusted for age and cohort for the discovery and validation datasets based on the CU mixed effect models. Volume/thickness per brain ROI was used as response, cohort, and subject id as random effects and age as a fixed effect in the CU mixed effect models (one model for each of the 41 left/right hemisphere averaged brain regions). Adding cohort as a random effect in these models enabled us to make individual average volume/thickness predictions for the effects of ADNI, J-ADNI, and AIBL cohorts and use the population mean that corresponds to all individuals to harmonize the data of the AddNeuroMed cohort. The addition of the cohort random effect at this step of the analysis, allows for future classification of MRI data from new cohorts to the identified longitudinal clusters. Adding age as a fixed effect allowed us to accurately estimate the anatomical changes in the 41 brain regions due to aging since the CU dataset consisted of amyloid-negative healthy controls with up to nine MRI visits and a CU diagnosis during the sum of their future follow-ups. The mean volume/thickness (mixed effect model atrophy expected fitted value for specific cohort and chronological age) at any age and the standard deviation of it (residual plus random effects standard deviation) were used to calculate w-values or AD patients. Consequently, w-values in our AD group (both discovery and validation datasets) reflect brain atrophy that is caused by the disease, free from the healthy aging anatomical features and cohort effects. To visually inspect this correction method, we employed a multivariate mixed effect model[30] and visualized the results. After this correction, the effect of disease is what remained in the AD dataset to be assessed.

In the second step, we applied an in-house pipeline for longitudinal clustering to the discovery dataset[15,31]. The multivariate mixture of generalized mixed effects clustering model[32] incorporates Bayesian inference to explore heterogeneity in the longitudinal brain data. We applied this model on brain volume/thickness but it has already been used in other applications[31,33–42]. The Bayesian approach allowed the implementation of a complex hierarchical model for each cluster, where probabilistic mixture modeling (number of clusters) is combined with mixed effect modeling (number of brain regions) in one model definition. The covariance between random effects (brain regions) for each cluster was also modeled and thus the outcome included structural and functional relationships between brain regions at a cluster level. This implementation allows for the inclusion of random and fixed effects. The feature of time (timescale) was modeled by random intercepts and slopes (cluster specific). Disease duration (time between AD onset age and MRI session age), was used as the timescale variable (cluster-specific random effect) in the clustering model. Since the existence of disease duration as a continuous measure in months was the initial inclusion criterion for AD patients in this study (for the design of the study), the timescale is common and disease duration exists for the sum of AD patients in the model. The fixed effects modeled the population level effects. MRI field strength and eTIV were used as fixed effects, assuming that they vary and follow a trend in the population but not between clusters. Following this approach, we accounted for some important effects that can bias the clustering algorithm[9]. Since the longitudinal clustering is based on the traditional model-based probabilistic clustering (mixture of Gaussian distributions), we also estimated patient-cluster probabilities that reflect the chances of a patient belonging to each cluster given their response vector (random intercepts and slopes)[43]. Patients with similar volume/ thickness patterns at the AD onset and progression over time were clustered together. Post-clustering, we assessed how many AD

volume/thickness patterns exist at the clinical AD dementia onset, the rate of atrophy per year for each pattern, and the frequency of each pattern in the population. Although internal measurements of clustering quality (Silhouette, CH, and others) already exist for clustering assessment, they do not provide enough information when longitudinal data are clustered. Instead, model-fitting information including model deviance and percentage of MCMCs with higher autocorrelation compared to the majority of MCMCs were used to assess clustering quality[15,31]. The model was initially fitted with linear slopes. After a sufficiently long simulation, the parameters were saved and then used to initialize the optimization again, but with the addition of quadratic terms. The second optimization step aimed to model the atrophy plateau that occurs after long disease duration. By following this stepwise quadratic term addition, we avoided the risk that all trend parameters (slopes and quadratic terms) are poorly optimized, which is common in models with as many parameters[44] as is our model-based clustering approach. Smoothed median-fitted value maps with volume/thickness at AD onset and for 8 consecutive years were calculated (using longitudinal information from patients at any time during the first 8 years of clinical disease duration) to characterize cluster volume and thickness tendency. Two different thresholds, 1.6 standard deviations below the CU normative values[45], and 0.5 standard deviations below the CU normative values (a less conservative threshold) were used for the atrophy maps.

In the third step we used the discovery set model as a classifier, to assess the chance of each patient in the validation dataset belonging to any of the defined clusters[46]. We used the validation dataset for two reasons. Since the validation dataset includes mainly patients with one MRI visit (79% of patients), we aimed to understand whether we can utilize the longitudinal model outcome with this cross-sectional information to accurately assign patients to the longitudinal clusters. To compare the accuracy of this assignment we calculated median volume/thickness images for the sum of patients in each cluster of the validation set separately. Then, we compared those median images with the fitted values (estimated at the median disease duration in months of the validation set for each cluster) of our model (2nd step) to make an approximate assessment of the classification ability of new AD patients' data. This helped us to increase the transparency of the supervised classification procedure and assess the model's ability to make relevant patient assignments into clusters. Moreover, by predicting cluster assignment in the validation dataset we were able to increase the size of the final clusters (pooled discovery and validation datasets) and make more accurate estimations of the cognitive profiles (and other characteristics) of the AD patient clusters. A further validation of the clustering method involved the application of the second step of the analysis independently in the ADNI and J-ADNI/AIBL datasets, to assess the volume/thickness patterns in the different datasets and their agreement to the complete dataset model. The correspondence between the results of the independent analysis in the ADNI and J-ADNI/AIBL datasets and their relation to the complete dataset analysis were assessed by means of distance between the intercepts and slopes of the identified patterns.

Some of the advantages of the overall pipeline are that it: incorporates whole brain data, leverages data of patients with different visit numbers and at different times, provides cluster visualization through the fitted values, provides clustering uncertainty measures, allows for the modeling of confounding effects, compares the patient's cluster specific volume/thickness with a group of healthy individuals[15], can potentially be used for the classification of new patients with only one MRI visit. In comparison to previous approaches[10,13], longitudinal data are used in longitudinal modeling and not as an evaluation set in cross-sectional analysis.

## Complementary statistical analysis

As mentioned in step one of the longitudinal clustering analysis, we estimated cluster-specific random effects covariance matrices for each cluster. Each element of the cluster-specific (one for each cluster) intercept covariance matrix represents the correlation of one brain region's intercept to any other region. Consequently, correlated brain regions may have similar structural connectivity. The same applies to slope covariance matrices. We are focusing more on slopes that can provide more information about structural connectivity. Thus, correlating random slopes shows that brain regions develop atrophy in a similar manner over time. It is important to notice that the intercept/slope variance/covariance matrices per cluster refer to estimated regression random intercepts and slopes and not to the original volume/thickness data[32]. To characterize the differences between clusters in terms of structural (intercept) and longitudinal (slope) brain regional volume/thickness relationships, nodal strength[47] was calculated based on the aforementioned intercept and slope variance/covariance matrices. This graph theory measurement summarizes information from covariance matrices for each brain region and reflects the sum of the correlations of a brain region with all the regions connected to it. Clusters were compared in pairs using BRAPH (http://braph.org/)[48]. It is important to stress that the nodal strength calculation was not used as the main analytical step in this study but only to help summarize the information from the cluster covariance matrices and to decrease the number of brain regions involved in the cluster interpretation. Moreover, post-clustering (after the main clustering analysis), the intercept and slope mean values per cluster were further clustered using hierarchical clustering, to investigate the existence of common atrophy intercepts and atrophy progression patterns (slopes) over time. This step helped to infer whether some clusters of patients follow the same spatial distribution of atrophy in the brain, but with faster or slower progression and/or different intercepts at the AD onset (stage of atrophy at the AD onset). For the ADAS-cog subscales, MMSE, and ANART, we applied generalized linear mixed effect models (and corrected our results post hoc) to explore differences between clusters. All analyses were done with R (3.6.3). ANART scores were used to assess premorbid intelligence.

## Reporting summary

Further information on research design is available in the Nature Research Reporting Summary linked to this article.

## Data availability

The datasets generated and analyzed during the current study are not available on their entirety due to individual agreements with the four cohort (ADNI, JADNI, AIBL, AddNeuroMed) committees. The datasets can be acquired after request to the individual cohort repositories. Unique deidentified ids of patients in each cluster and the clustering results and full models outputs can be shared upon reasonable request. TheHiveDB was used for processing of images with Freesurfer 6.0.0.

## Code availability

All relevant code is included in the supplementary file: Supplementary Software 1.

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

## Acknowledgements

We would like to thank participants of J-ADNI and ADNI, AIBL, AddNeuroMed studies and their family members who made this study possible. We thank J-ADNI and ADNI, AIBL, AddNeuroMed colleagues for their contributions to the work summarized here. Moreover, we would like to thank the Swedish Foundation for Strategic Research (SSF), The Swedish Research Council (VR), the Center for Innovative Medicine (CIMED), the Strategic Research Programme in Neuroscience at Karolinska Institutet (StratNeuro), Swedish Brain Power, the regional agreement on medical training and clinical research (ALF) between Stockholm County Council and Karolinska Institutet, the joint research funds of KTH Royal Institute of Technology and Stockholm County Council (HMT), Hjärnfonden, Alzheimerfonden, the Åke Wiberg Foundation, and Birgitta och Sten Westerberg for additional financial support. Data collection and sharing for this project were funded by the Alzheimer's Disease Neuroimaging Initiative (ADNI) (National Institutes of Health Grant U01 AG024904) and DOD ADNI (Department of Defense award number W81XWH-12-2-0012). ADNI is funded by the National Institute on Aging, the National Institute of Biomedical Imaging and Bioengineering, and through generous contributions from the following: Alzheimer's Association; Alzheimer's Drug Discovery Foundation; BioClinica, Inc.; Biogen Idec Inc.; Bristol-Myers Squibb Company; Eisai Inc.; Elan Pharmaceuticals, Inc.; Eli Lilly and Company; F. Hoffmann-La Roche Ltd and its affiliated company Genentech, Inc.; GE Healthcare; Innogenetics, N.V.; IXICO Ltd.; Janssen Alzheimer Immunotherapy Research & Development, LLC.; Johnson & Johnson Pharmaceutical Research & Development LLC.; Medpace, Inc.; Merck & Co., Inc.; Meso Scale Diagnostics, LLC.; NeuroRx Research; Novartis Pharmaceuticals Corporation; Pfizer Inc.; Piramal Imaging; Servier; Synarc Inc.; and Takeda Pharmaceutical Company. The Canadian Institutes of Health Research is providing funds to support ADNI clinical sites in Canada. Private sector contributions are facilitated by the Foundation for the National Institutes of Health (www.fnih.org). The grantee organization is the Northern California Institute for Research and Education, and the study is coordinated by the Alzheimer's Disease Cooperative Study at the University of California, San Diego. ADNI data are disseminated by the Laboratory for Neuro Imaging at the University of California, Los Angeles. Data used in preparation of this article were obtained from the Alzheimer's Disease Neuroimaging Initiative (ADNI) database (adni.loni.ucla.edu). As such, the investigators within the ADNI contributed to the design and implementation of ADNI and/or provided data but did not participate in the analysis or writing of this report. A complete listing of ADNI investigators can be found at: http://adni.loni.usc.edu/wp-content/uploads/how_to_apply/ADNI_Acknowledgement_List.pdf.

## Author contributions

K.P.: conceptualization, data curation, formal analysis, investigation, methodology, project administration, validation, visualization, writing original draft. J.B.P.: conceptualization, investigation, formal analysis, visualization, writing original draft, writing—review & editing. J-S.M.: data curation, writing—review & editing. L.-O.W.: conceptualization, writing—review & editing. Ö.S.: writing original draft, writing—review & editing. G.V.: writing original draft, writing—review & editing. C.M.: conceptualization, writing—review & editing. D.A.: writing original draft, writing—review & editing. J.N.: writing—review & editing. T.I.: writing original draft, writing—review & editing. D.F.: conceptualization, investigation, writing original draft, writing—review & editing. E.W.: conceptualization, investigation, supervision, writing original draft, writing—review & editing, project administration, and resources. Japanese Alzheimer's Disease Neuroimaging Initiative: data gathering and quality control. Australian Imaging, Biomarkers and Lifestyle study: data gathering and quality control.

## Funding

## Competing interests

The authors declare no competing interests.

## Additional information

## Japanese Alzheimer's Disease Neuroimaging Initiative

Takeshi Iwatsubo[8]

## Australian Imaging, Biomarkers and Lifestyle study

Colin L. Masters[5] & David Ames[6,7]

Lists of members and their affiliations appears in the Supplementary Information.

