## [Peer Review File · Nature Communications]

Multi-cohort and longitudinal Bayesian clustering study of stage and subtype in Alzheimer's diseaseReviewers' Comments:

Reviewer #1:

Remarks to the Author:

This study presents an elegant approach to determine atrophy subtypes in individuals with Alzheimer's disease dementia, based on longitudinal data. The combined data set is large, and the availability of repeated MRI and cognition as well as another validation dataset that was not part of the clustering solution is another strong aspect of this study.

Some questions:

-For the discovery dataset, it is important to understand the influence of the different datasets on the solution, even though much care has been put in to harmonise the data. It would be highly informative if the authors could show what clustering solutions look like when the methodology is applied for the separate cohorts, as well as a confusion matrix whether individuals would be labelled to belonging to the same cluster as that presented in the main analyses.

-For the influence of disease severity, it would be of interest to repeat clustering for a subset of AD dementia patients restricted to a narrow disease severity range, for example, having an MMSE score between 24-20: Does this still result in a 'severe' cluster?

-It is remarkable that the largest cluster did not show atrophy in AD patients compared to controls, since atrophy is one of the stronger correlates of disease severity (i.e., having dementia): please explain. Also, another check, when comparing hippocampal volume, did this also not differ from controls? Might this perhaps also reflect the scaling step that was performed (p6 mentions a 'Z-score' possible the atrophy coefficients were normalised across the AD group, and then 0 would indicate average atrophy, instead of no atrophy...?).

-For the validation cohort, were there individuals who did not really fit into any of the clusters? Since no amyloid information is available for all of those individuals, it might be that those without amyloid may not fit to AD subtypes. Or, if they do, it may mean that the atrophy patterns are not specific to AD perse.

Reviewer #2:

Remarks to the Author:

The manuscript by Poulakis et al. used longitudinal MRI data in amyloid positive individuals with an Alzheimer's disease diagnosis to cluster individuals with similar cross-sectional and longitudinal atrophy patterns. The authors identify five clusters, which they summarise to represent different stages of two pathways of neurodegeneration.

Characterising heterogeneity in longitudinal atrophy patterns is an important topic and main the strength of the work is the use of longitudinal data from multiple cohorts. However, the major limitation of the analysis is that the timespan of the longitudinal data is very short with respect to the full (decades-long) disease duration. In particular, the authors only consider individuals with a diagnosis of Alzheimer's disease and so are unable to assess atrophy in the long pre-symptomatic portion of the disease. There are also very few longitudinal follow-up visits past the 4th time point (as shown in Table S1), which is only a median of 1.5 years from baseline, and only two subjects with follow-up visits past the 5th time point, which is still only a median of 2.5 years from baseline.

Further comments:

- The authors state that in the literature "current findings are based on cross-sectional analyses", that "longitudinal information was not used" and that "Assuming individuals will remain in the same

subtype using one observation, without longitudinal evidence is questionable.” It is true that many studies train their models on cross-sectional data, but many studies also subsequently validate on longitudinal data and demonstrate stability of their clustering at follow-up. Examples include Zhang et al. PNAS 2016, Vogel et al. Nature Medicine 2021.

- Several points in the manuscript refer to measures of “intelligence”. Is this meant to mean the education score? References to intelligence should be changed to education if so.
- The rationale for choosing the number of clusters is not well explained. In particular, there is no reference for using the percentage of converged MCMC parameters as a measure of clustering quality. Typically, a lack of convergence of the MCMC indicates that the optimal parameters have not been found, not that the quality of the clustering is worse. This means that the authors need to fix the MCMC chain, not select a different clustering solution.
- The logic behind initially fitting the model with linear atrophy slopes and then adding in quadratic terms for the last iterations was unclear to me. Why not model both at once for all simulations? Why not perform model selection to determine whether to include a quadratic term?
- Throughout the manuscript the authors refer to there being 8 years worth of data and they fit the model for 8 years, but in Table S1 the duration of follow-up seems to be considerably less for most individuals (see major point above). Where did the figure of 8 years come from?
- I was not convinced that the results provide enough evidence to support the conclusion there are two pathways of neurodegeneration. First, only AD subjects are included and so a large portion of the atrophy trajectories is missed. Particularly in individuals with cortical and diffuse atrophy, there is no way of inferring the trajectory they underwent to get to that point. Second, there is no statistical comparison of the regional rates of atrophy in the different groups.
- The validation experiment seems quite weak. If you assign individuals to clusters based on their atrophy pattern shouldn't they always express the atrophy pattern of that cluster? The rationale behind this experiment needs further explanation. It would be a much stronger validation to cluster the validation set separately and demonstrate the clusters are the same. Also there is no real longitudinal validation – the validation dataset has a median total follow-up duration of 1 year. Why are the amyloid negatives included in the validation dataset? Are there any differences in cluster assignment and characteristics between the amyloid positives and negatives?
- Is demonstrating that atrophy patterns converge in late stages a novel finding? Seems to be inevitable given that more regions of the brain atrophy with disease progression?

Reviewer #3:

Remarks to the Author:

Overview of Analysis:

The authors take on the issue of determining whether putative AD subtypes observed in previous studies are actually intermediate stages in more generalized disease trajectories. They make the observation that previous cross-sectional neuroimaging papers are inherently unable to account for differences in disease trajectory, and thus use an longitudinal cohort of patients from 4 separate datasets in order to explore/cluster longitudinal changes in brain structure. They use patients w/ amyloid-confirmed AD from populations on 4 separate continents, and place all of them on a common timescale beginning at the time of AD diagnosis.

The authors first construct a multilevel model of normal age-related brain atrophy for each cohort. They accomplish this using amyloid-negative, cognitively unimpaired patients from the datasets in

order to provide baseline atrophy levels to later compare changes observed in AD patients. Each brain region (both cortical and subcortical, derived from the Desikan atlas) is modeled for atrophy over time using cohort _ subject as random effects, and age as the sole fixed effect.

The authors then apply a clustering model that groups patients by the atrophy dynamics measured in each brain region over time. Specifically, their approach treats each brain region as a separate response variable, with associated random slopes and intercepts. The intercepts represent the atrophy in a region at the time of diagnosis, whereas the slopes represent the speed of atrophy per region over the study period.

In this case, atrophy is as measured by something called the w-score. This is essentially a Z-score re-adjusted for age and cohort.

The w-score metric is derived in relation to the normal degree of age-related atrophy that the authors estimate in their initial multilevel model of cognitively-unimpaired patients.

The clustering model assumes a mixture of hierarchical models for w-scored atrophy. Random and fixed effects are fitted by Markov Chain Monte Carlo sampling, thus making the analysis Bayesian in nature (hence their title).

Disease duration (from time of AD diagnosis) was used as a cluster-specific random effect.

Intracranial volume and MRI field strength were used as population-wide fixed effects.

By utilizing w-scoring and hierarchical modeling, the authors argue that they are sufficiently able to disentangle pathologic changes from normal aging. Furthermore, by starting their modeling at the time of AD diagnosis, they argue that all patients are analyzed on a consistent timescale.

The authors test this mixture of hierarchical models using cluster sizes between 2 and 8. They ultimately state that 2- and 5-cluster solutions are superior solutions, as they minimize model deviation and featured the greatest degree of parameter convergence during MCMC sampling.

They ultimately decided to base further analyses on the 5-cluster model, it seems (they come up with names for these clusters later in the paper).

Finally, the authors turn their hierarchical model for AD atrophy into a classifier, and use this on a validation set of patients from each dataset. They population-average the model's predictions at the median disease duration, and then visualize the output by translating the predictions back to an "average" cortical surface per cluster. They then assess goodness of prediction by visually-comparing to an average cortical surface taken from the sum of observations at the median disease duration.

Of note, the authors also conducted 2 complementary analyses:

They translated cluster-specific covariance matrices (ie covariance between random effect slopes and intercepts in each brain region) to graphs in which brain regions are nodes and pairwise covariances are edges. They used something called "nodal strength" in order to proxy the connectedness of atrophic changes each cluster, and then performed pairwise comparisons or nodal strength for different clusters.

They also performed multilevel modeling to compare differences in neurocognitive testing scores between clusters.

Findings/Implications:

-- The authors ultimately found that the 5 separate clusters they identified in their modeling had differing degrees of brain atrophy at the time of diagnosis, as well as the speed of atrophy observed across subsequent visits.

However, by visual inspection of their model outputs, they ultimately concluded that all 5 clusters fell into one of two more general patterns of structural degeneration: mediotemporal vs. cortical spreading.

-- Various clusters seemed to converge at later time points in the disease course, based on the ultimate pattern of atrophy observed at the end-points of the study.

Differing clusters had differing results on neurocognitive testing. Interestingly, patients with greater degrees of atrophy at time of diagnosis had greater premorbid intelligence, suggesting a protective effect from "cognitive reserve."

Differing clusters had different degrees of nodal strength.

-- The ultimate point here (in my opinion): Though differing clusters were successfully derived from the datasets based on the dynamics of their longitudinal atrophy patterns, they ultimately did converge to two main pathways of structural change. This suggests that imaging subtypes previously observed on cross-sectional studies may be but intermediate stages in much more general patterns of structural change.

Strengths:

-- Generally, I believe that the complexity of the modelling appropriately matches the complexity of the problem. Mixed effects modelling with an overlying clustering approach seems appropriate to account for subject-level vs. population-level effects, while also accounting for the unresolved questions in the field regarding the existence of AD subtypes.

-- The data sources are very strong: 4 cohorts from 4 different continents, all with similar protocols for data acquisition and data harmonization.

-- I feel that the conclusions are made in the discussion are appropriate and quite feasible; furthermore, if this is indeed the first longitudinal imaging study of its kind, it is a very important contribution to the field.

Weaknesses:

-- The writing is somewhat inelegant in places. Though I realize this is an international group of researchers, I am wondering if a bit more could be done to smooth it in spots. The writing is superfluous in some spots, or does not go into sufficient detail in others.

-- There is no mention of the patients' comorbidities and how these were/were not accounted for in the analysis. This is important, as the authors speculate that non-AD pathologies may play a role in the observed changes, but they never elaborate on it.

-- The graph theory analysis needs more in-depth explanation. I am wondering if it is too much of a jump to suggest that stronger covariances from the modelling are automatically suggestive of structural correlation (at the very least, it is a claim that could use additional proof or analysis).

-- I am also a bit confused what their implication is in comparing nodal strength between clusters. Perhaps that pathologic changes are more strongly propagated in certain of the clusters? I would be interested if they could back-correlate this with the trajectory of pathologic changes observed in some way.

-- It is not entirely clear to me why the 5-cluster solution was ultimately chosen over the 2-cluster solution. The authors tout them as being equally good, but ultimately elect to discuss the 5-cluster solution for the majority of the paper.

-- Do the authors provide a quantitative basis for their claim that the 5 clusters ultimately boil down into cortical vs. mediotemporal spreading categories? Or is it just visual?

In sum, I think this paper should be considered for publication with major revisions.

REVIEWER COMMENTS

#Beginning of reviewers' comments

Reviewer #1 (Remarks to the Author): This study presents an elegant approach to determine atrophy subtypes in individuals with Alzheimer's disease dementia, based on longitudinal data. The combined data set is large, and the availability of repeated MRI and cognition as well as another validation dataset that was not part of the clustering solution is another strong aspect of this study.

Some questions:

Comment_1

-For the discovery dataset, it is important to understand the influence of the different datasets on the solution, even though much care has been put in to harmonise the data. It would be highly informative if the authors could show what clustering solutions look like when the methodology is applied for the separate cohorts, as well as a confusion matrix whether individuals would be labelled to belonging to the same cluster as that presented in the main analyses.

Answer to the comment:

We believe this is a very relevant comment. Applying clustering in the different cohorts to assess whether all subtypes exist in all the sub populations would shed light on the heterogeneity within each cohort separately. Moreover, this approach would help us to evaluate how stable each clustering solution is when sample changes are introduced. However, we decided to follow a cohort "correction" approach because we identified one sample issue and one statistical issue when the cohorts were analysed separately. For instance, as it can be observed in table 2, the sample sizes of each cohort are very different, increasing the chances of finding more clusters in a bigger sample. To answer this sample size limitation, we combined the samples to obtain a larger and more heterogeneous sample after correcting statistically for cohort differences. The statistical issue with the separate cohort analyses is that a statistical model applied to the ADNI cohort will be much stronger than a statistical model applied to the AIBL cohort because the number of parameters will be the same (fixed and random effects), while the number of observations will not be similar. This will not enable us to compare the two models in statistical terms. Moreover, the model optimized for AIBL or J-ADNI will not be a statistically good model because the parameters exceed by far the number of observations. Studies on the heterogeneity of AD have been using this approach (separate cohort analyses), while disregarding this limitation. In our opinion if one model is much "weaker" than another in the same analyses the chances for reproducibility decrease radically. To sum up, we decided to train one model in the main analysis of this study, in order to build one robust statistical model rather than many weak models. We are happy to discuss further our approach with the reviewer if we have missed some relevant point in our decision. We have now commented on our choice of a single model in the discussion section (last page of discussion/page 15 of the manuscript).

Comment_2

-For the influence of disease severity, it would be of interest to repeat clustering for a subset of AD dementia patients restricted to a narrow disease severity range, for

example, having an MMSE score between 24-20: Does this still result in a 'severe' cluster?

Answer to the comment:

We understand the reviewers concern. We accounted for it in the study design by modelling time from the clinical disease onset. As the reviewer points out, we could exclude cases with MMSE equal/lower to 20 in a complementary analysis. The clusters would have been the same, but we would have missed the patients that help to model to cover the late stages of each cluster's atrophy. This claim is supported by the fact that the MMSE scores at baseline reported in Figure 3 (all five patient clusters have AD onset estimated MMSE score greater than 21, but the average values reduce per year of disease progression). Also, the range of MMSE at the baseline MRI visit (baseline is not the value at AD onset) is between 9 and 29 but the mass of the distribution (majority of individuals, almost 95% of the cases) takes values around 23 (median value, see figure below).

To conclude, since we modelled disease severity and heterogeneity simultaneously, our aim was to include patients at later stages of the disease and with lower MMSE scores to "model" the progression of each atrophy cluster. To address the reviewer's concern, we have now added the median, 1st and 3rd quartile values of MMSE at baseline MRI visit in the legend of table 2.

Comment_3

-It is remarkable that the largest cluster did not show atrophy in AD patients compared to controls, since atrophy is one of the stronger correlates of disease severity (i.e., having dementia): please explain. Also, another check, when comparing hippocampal volume, did this also not differ from controls? Might this perhaps also reflect the scaling step that was performed (p6 mentions a 'Z-score' possible the atrophy coefficients were normalised across the AD group, and then 0 would indicate average atrophy, instead of

no

atrophy...?).

Answer to the comment:

We agree with the Reviewer that it is indeed remarkable but we also believe that, from a methodological perspective, it is expected that the largest group of patients does not show extensive atrophy that would be 1,6 SD below the CU group at the clinical AD onset since this group gradually developed atrophy in the temporal lobe over time. Since the scaling was done with respect to the CU group, there is no chance that the value 0 reflects average atrophy at the patient group level. However, in the manuscript we state that the images show fitted values (in colour) when the values exceed a certain threshold. By doing this, one of the reasons why the mediotemporal lobe did not show earlier atrophy (indicated by yellower colours) is also the cut off atrophy value. By so, we mean that in our study we wanted to be conservative and chose to consider for visualization only the fitted values below 1,6 standard deviations from the CU group. We would like to clarify that we do not claim that the MA group has no atrophy but that it shows no large variations from the group of CU individuals, after controlling for age, intracranial volume, and MRI field strength at the AD onset. We agree with the reviewer that the lack of hippocampal atrophy reflects the scaling process, but this is because the difference is not as obvious early in the clinical course of the disease as in more advanced stages. In the systematic review of Ferreira and colleagues (Ferreira et al 2020), it is mentioned that MA is a large group of patients that is found in the studies that assess subtypes of AD. Moreover, since in our study we estimate for the first time longitudinal patterns of atrophy for the different AD subtypes from the AD onset, our MA pattern includes patients both from the minimal atrophy, limbic predominant subtypes, and potentially also subjects from the early stages of the typical AD subtype of atrophy found in the literature . Moreover, we only call this cluster MA because it shows minimal atrophy at the AD onset. This name was selected to help the readership to compare our results with the literature where the minimal atrophy subtype of atrophy is systematically identified (Ferreira et al 2020). Later in the disease course, all patients of this large cluster develop the mediotemporal atrophy pattern.

One thing that is important to mention regarding the MA cluster of patients is that it reflects the most heterogeneous group in the results, showing a high within group variability. In clustering analyses, it is common that the algorithm identifies some clusters of observations that are very tight (show low within cluster variability) and one cluster that is more heterogeneous than the others and gathers the most representative observations of the sample (for additional information on this clustering phenomenon please see Poulakis et al 2018). This is the case of the cluster that we call MA. In a future study we aim to identify sub-groups of patients within this cluster and understand whether there are factors other than atrophy that can explain their variability (glucose metabolism, tau spread etc).

We have now added a complementary discussion on the MA pattern of atrophy at page 12 of the manuscript.

Poulakis, Konstantinos, et al. "Heterogeneous patterns of brain atrophy in Alzheimer's disease." *Neurobiology of aging* 65 (2018): 98-108.

Ferreira, Daniel, Agneta Nordberg, and Eric Westman. "Biological subtypes of Alzheimer disease: A systematic review and meta-analysis." *Neurology* 94.10 (2020): 436-448.

Comment_4

-For the validation cohort, were there individuals who did not really fit into any of the clusters? Since no amyloid information is available for all of those individuals, it might be that those without amyloid may not fit to AD subtypes. Or, if they do, it may mean that the atrophy patterns are not specific to AD perse.

Answer to the comment:

We would like to thank the reviewer for this excellent comment. We added in the original manuscript the first and second order assignments (first order: most probable cluster, second order: second most probable cluster for individuals that do not belong certainly to any of the clusters) for the discovery dataset (see table S4). Now, we estimated also the second order assignments for the validation cohort and added them in the supplementary material. The results in the validation dataset are added on page 10 of the manuscript (results, Atrophy patterns, discriminant features, and model validation) and in the supplementary material (table S5 of the supplementary material). Only 5 out of 571 patients had an uncertain clustering assignment and it did not relate to amyloid beta status. Also, we added a patient cluster probability plot (Figure S6), which is a visual guide of the results mentioned in table S5 but also shows the relationship between all five clusters in terms of classification proximity.

Reviewer #2 (Remarks to the Author):

The manuscript by Poulakis et al. used longitudinal MRI data in amyloid positive individuals with an Alzheimer's disease diagnosis to cluster individuals with similar cross-sectional and longitudinal atrophy patterns. The authors identify five clusters, which they summarise to represent different stages of two pathways of neurodegeneration.

Comment_5

Characterising heterogeneity in longitudinal atrophy patterns is an important topic and main the strength of the work is the use of longitudinal data from multiple cohorts. However, the major limitation of the analysis is that the timespan of the longitudinal data is very short with respect to the full (decades-long) disease duration. In particular, the authors only consider individuals with a diagnosis of Alzheimer's disease and so are unable to assess atrophy in the long pre-symptomatic portion of the disease. There are also very few longitudinal follow-up visits past the 4th time point (as shown in Table S1), which is only a median of 1.5 years from baseline, and only two subjects with follow-up visits past the 5th time point, which is still only a median of 2.5 years from baseline.

Answer to the comment:

We would like to take the opportunity to stress that the reviewer is right regarding the pre-symptomatic portion of the disease which is not modelled in this study. In our study, we explicitly aimed to model the dementia stage of the disease where most of the

atrophy changes occur. However, we have now added the fact that we do not model the pre-dementia stage as a limitation on page 16.

The number of follow up MRI visits for each patient are not many as the reviewer accurately points out. However, it is important to mention here regarding the benefit of our analysis, that although the timepoints are not many, we modelled trajectories of decline with slopes and therefore an interval of 1.5 years is enough to observe how atrophy develops. Of course, we agree with the reviewer that a longer follow up will increase the certainty of the patient ROI atrophy slope estimation. Since ADNI and AIBL continue their follow up visits one can try to re estimate those slopes when even more data are available. We have added the absence of long follow ups as a limitation to the discussion section, page 15 and 16.

Further comments:

Comment_6

- The authors state that in the literature “current findings are based on cross-sectional analyses”, that “longitudinal information was not used” and that “Assuming individuals will remain in the same subtype using one observation, without longitudinal evidence is questionable.” It is true that many studies train their models on cross-sectional data, but many studies also subsequently validate on longitudinal data and demonstrate stability of their clustering at follow-up. Examples include Zhang et al. PNAS 2016, Vogel et al. Nature Medicine 2021.

Answer to the comment:

We agree with the reviewer that the seminal study of Zhang and colleagues (with latent Dirichlet allocation modelling) and the Sustain model applied in Tau imaging data from Vogel and colleagues used longitudinal data for validation of their models. However, the distinction between those studies and ours is that they estimated patterns of disease only at one specific time point and then checked whether individuals changed cluster over time, while in our study we show how each pattern or cluster of atrophy evolves over time. In our approach, longitudinal data are used in the modelling while in the other mentioned studies they are only used for validation. Thus, our study is the first in implementing these methodological developments, which allow to assess how the clusters change over time. We realized that this methodological difference was not stressed. We have added a comment regarding this it in the methods section (longitudinal clustering analysis) to further stress the novelty of our study, page 8.

Comment_7

- Several points in the manuscript refer to measures of “intelligence”. Is this meant to mean the education score? References to intelligence should be changed to education if so.

Answer to the comment:

We apologize for the confusion. We would like to thank the reviewer for pointing this out. We had two measures of cognitive reserve available from the four cohorts that were used in the study. We used years of education (see page 13) and the ANART scale (American national adult reading test), which is a common measure to estimate

premorbid intelligence. When in the manuscript we mention the word intelligence we refer to the ANART scale (see page 4, end of introduction). We have clarified this on page 9 (complementary statistical analysis).

Comment_8

- The rationale for choosing the number of clusters is not well explained. In particular, there is no reference for using the percentage of converged MCMC parameters as a measure of clustering quality. Typically, a lack of convergence of the MCMC indicates that the optimal parameters have not been found, not that the quality of the clustering is worse. This means that the authors need to fix the MCMC chain, not select a different clustering solution.

Answer to the comment:

This is a very important comment. The reviewer is right, we failed to explain our reasoning well in the manuscript. We have now clarified how we used the MCMC metrics. The updated description of the model assessment can be found on page 7 of the manuscript and in the legend of table S3. When we mentioned in the manuscript the convergence of an MCMC chain we do not mean in any case that any of the chains has not converged. We used the MCMC autocorrelation coefficient to have a numerical value of convergence and calculated the percentage of chains with coefficient lower than 0.9. Of course, there were no chains that failed to converge or lagged in the optimization process. On a related note, optimization processes such as this clustering model, that includes hundreds of chains/parameters can have some chains that converge slower than other ones, often due to the lack of information in a specific variable. As an example, if only one ROI in the brain is not clusterable while the rest are, then there will be no separate intercepts or slopes between clusters (they will all have the same intercepts and slopes). Those intercepts and slopes may show slower convergence/higher autocorrelation than a clearly clusterable ROI. This does not produce a solution that cannot be used for interpretation but may help to separate a good from a better model. We used this approach on our published methodological paper (see Poulakis, et al 2020).

Poulakis, Konstantinos, et al. "Fully Bayesian longitudinal unsupervised learning for the assessment and visualization of AD heterogeneity and progression." *Aging (Albany NY)* 12.13 (2020): 12622.

Comment_9

- The logic behind initially fitting the model with linear atrophy slopes and then adding in quadratic terms for the last iterations was unclear to me. Why not model both at once for all simulations? Why not perform model selection to determine whether to include a quadratic term?

Answer to the comment:

We understand the question about whether it would be better to start the model training with all the model parameters (linear and quadratic terms). As it is discussed in comment 5, there are many follow-up visits for individuals in the discovery dataset are many. However, we do not believe that they are enough to train a complete model with quadratic terms since the observations/parameter number balance will be very weak for models with many clusters (6-8 clusters). Instead of trying to overfit the models to our

discovery dataset, we chose to first build linear models that will have robust slope parameters. When those parameters were trained based on the discovery dataset's likelihood information, we restarted the model optimization using the already existing near optimal parameter values as starting values for the slopes and as quadratic terms the value 0. In this way the slopes may change slightly just to "fit" the added slope information due to the existence of the new quadratic term. In this stepwise quadratic term addition, we avoided the chance that all trend parameters (slopes and quadratic terms) are poorly optimized, which is common in models with as many parameters as is our model-based clustering approach. We agree with the reviewer that a more complete solution is to build full models (intercept, linear, and quadratic term) from the beginning. However, we believe that even more data are needed to obtain a statistically robust parameter estimation in such quadratic models. This is why we used the stepwise quadratic terms addition. We have now addressed and clarified this in the methods section of the manuscript, page 7 (longitudinal clustering analysis).

Comment_10

- Throughout the manuscript the authors refer to there being 8 years worth of data and they fit the model for 8 years, but in Table S1 the duration of follow-up seems to be considerably less for most individuals (see major point above). Where did the figure of 8 years come from?

Answer to the comment:

We apologize for the misunderstanding. We have now clarified this in the manuscript (Methods, longitudinal clustering, page 7) so that the "8 years worth of data" is explained more clearly. Model-based clustering estimates cluster characteristics for a certain span of "time". We chose to model 8 years after the clinical AD onset. To do so, an intercept, a slope, and dispersion parameters were estimated. Using those parameters, we estimated the patterns of atrophy for each cluster from the AD onset and for the next 8 years. The data that were used to do so did not consist of patient's information over 8 years. The advantage of our method is that any longitudinal information at some point within this span (8 years) or afterwards was used to reconstruct the 8 years trajectory of atrophy. As an example, one patient may have MRI information between the 2nd and 5th year after the AD onset while another patient may have information for the first three years after the AD onset. If their trajectories match, then those two patients will belong to the same cluster and the atrophy trajectory of the cluster that they belong to will be estimated with the likelihood information of the individual atrophy trajectories of the two patients. Below, please see attached an individual profile plot for the hippocampal volume reduction as a function of disease duration for a subset of individuals. One can see that each individual's trajectory of volume decline is spread somewhere in the disease duration scale. A cluster's trajectory over 8 years is an average line that "connects" all those individual trajectories and reflects a reconstruction of the average patient decline for that cluster over 8 years.

Comment_11

- I was not convinced that the results provide enough evidence to support the conclusion there are two pathways of neurodegeneration. First, only AD subjects are included and so a large portion of the atrophy trajectories is missed. Particularly in individuals with cortical and diffuse atrophy, there is no way of inferring the trajectory they underwent to get to that point. Second, there is no statistical comparison of the regional rates of atrophy in the different groups.

Answer to the comment:

We agree with the reviewer on the fact that data before the clinical AD dementia stage are not included in this study and therefore a large preclinical part of the disease is not modelled. However, this was intentional since in this study we aimed to model only the dementia stage of the disease. We are planning to obtain and combine CU, MCI, and AD data to model as much of the AD continuum heterogeneity as possible in the future (please, see also our answer to the 5th comment). However, our future study which will include patient data from the preclinical until the post mortem stage, will come with the price of reduced numbers of individuals since long preclinical follow ups of future AD patients are seldom available apart from the familial cases.

Regarding the diffuse atrophy cluster of patients, we agree with the reviewer. In the discussion section, page 13, we acknowledge that DA is a challenging cluster to interpret: *“To explain the atrophy trajectories of our DA cluster is challenging since excessive frontal and temporal atrophy was already present at the clinical onset.”* As for the cortical atrophy it is clear that patients that belong to that cluster have no prominent hippocampal atrophy before the diagnosis of AD. Cortical atrophy is localized to the lateral parietal lobe at the dementia diagnosis stage (figure 1) and before that there is no large differences between the CU group of individuals and potentially MCI’s that will develop AD of that type. We have now added some discussion about this in the limitations section (page 16).

Statistical comparisons of local atrophy rates between the different AD patient groups would have been the proper way to compare them if clustering did not precede the atrophy fitted values maps (see Whitwell et al 2012). However, since clustering is a statistical way of finding distinct groups of observations, its results are ready to be visualized. Instead of comparing the groups again and run into circularity issues that also increase the number of statistical comparisons in our study (and reduce the reproducibility), we chose to look for significant variables (variables that are profoundly discriminants between clusters) (see Appendix S4). In this way we compared groups of patients only one time (during clustering) and inferred statistically significant differences in atrophy through variable importance. Also, if the data were cross sectional (only one visit per patient), statistical comparisons between groups would have been straight forward (as we did in Poulakis et al 2018). Since the longitudinal data in our dataset are irregularly sampled (MRI visits did not take place at the same time with respect to the disease onset), it is very hard to make a fair comparison between groups of patients (in terms of regional atrophy rates) due to different disease stages.

Whitwell, Jennifer, et al. "Neuroimaging Correlates of Pathologically-Defined Atypical Alzheimer's Disease (P05. 049)." (2012): P05-049.

Poulakis, Konstantinos, et al. "Heterogeneous patterns of brain atrophy in Alzheimer's disease." *Neurobiology of aging* 65 (2018): 98-108.

Comment_12

- The validation experiment seems quite weak. If you assign individuals to clusters based on their atrophy pattern shouldn't they always express the atrophy pattern of that cluster? The rationale behind this experiment needs further explanation. It would be a much stronger validation to cluster the validation set separately and demonstrate the clusters are the same. Also there is no real longitudinal validation – the validation dataset has a median total follow-up duration of 1 year. Why are the amyloid negatives included in the validation dataset? Are there any differences in cluster assignment and characteristics between the amyloid positives and negatives?

Answer to the comment:

The reviewer is correct that if you assign individuals to pre-existing clusters, they will always express those clusters only. We know that the reasoning behind this validation strategy may be confusing at first glance and we have now explained better the reasoning behind our strategy in the methods section (page 8, methods section). We used a sample from each cohort (ADNI, JADNI, AIBL) and one new cohort that is "unknown" (AddNeuroMed) to the clustering model to compile the validation dataset. The aim with this dataset was not to reproduce the clusters of the discovery dataset. Our aims were to:

- 1) Validate whether patient MRIs that are classified through our classification model have atrophy rates that are similar to the ones that the actual clustering model has as fitted values at any estimated time from the AD dementia onset (see figure 2 of the manuscript). This is extremely important if we want to claim that our model is not a black box model, but it makes a sensible cluster choice. In this way we can claim

that it can be used to assign any new AD patient's MRI into one of the discovered clusters.

- 2) Assign as much data as possible to each cluster so that our conclusions about cluster characteristics (other than atrophy from MRI), such as demographics, cognition, genetic, and other, are estimated with higher accuracy (greater chances for generalizability and reproducibility) through the final combination of the discovery and validation dataset.

Also, running clustering on the validation dataset to see if the same clusters exist would be a better strategy to validate the clusters found in the discovery dataset. Therefore, we agree with the reviewer on this point too. However, most of the patients had only one image. It was our complementary goal (associated to our second main aim from above) to find out whether our longitudinal model can be used to classify cross-sectional data well.

Our discovery cohort which is the basis of the estimated atrophy patterns in AD included only amyloid positive AD patients. Patients that had unknown amyloid status or negative amyloid status were added in the validation dataset only. In our validation dataset we included the AddNeuroMed cohort since it is an additional source of good quality MRIs from AD diagnosed patients. In that cohort, amyloid beta status was unfortunately not available. Since there was a chance for each patient of that cohort to be either amyloid beta positive or negative, the exclusion of patients with unknown or negative amyloid status from the ADNI, JADNI, or AIBL cohorts in the validation dataset would introduce a bias in the validation dataset. This is the only reason why patients with unknown amyloid beta status were included in the validation dataset. No differences in cluster frequencies were observed between amyloid negative and positive AD patients. We have now added this result in the section cluster characteristics (page 11). Another new study that utilized data from amyloid negative individuals in the context of AD is the recent study from Vogel and colleagues in Nature medicine. Finally, we agree with the reviewer that even more longitudinal data would have allowed an optimal longitudinal validation of our results. We do not have it currently and it is our aim to gather more longitudinal data for longitudinal validations in the future.

Vogel, Jacob W., et al. "Four distinct trajectories of tau deposition identified in Alzheimer's disease." *Nature Medicine* 27.5 (2021): 871-881.

Poulakis, Konstantinos, et al. "Heterogeneous patterns of brain atrophy in Alzheimer's disease." *Neurobiology of aging* 65 (2018): 98-108.

Comment_13

- Is demonstrating that atrophy patterns converge in late stages a novel finding? Seems to be inevitable given that more regions of the brain atrophy with disease progression?

Answer to the comment:

We agree with the reviewer that it seems that it is inevitable for the atrophy patterns in AD to converge in late disease stages since more brain regions are involved in that process. We mention in the manuscript that this is a novel finding because the existing literature including a study of the first author of this manuscript (Poulakis et al 2018) and Vogel and colleagues (Vogel et al, 2021) discuss all subtypes as distinct entities (they refer to Tau and not atrophy). This happens probably due to the cross-sectional nature of

those studies. In this study, we demonstrate the convergence of patterns based on longitudinal MRI assessments. We completely agree with the reviewer that this development is expected.

Vogel, Jacob W., et al. "Four distinct trajectories of tau deposition identified in Alzheimer's disease." *Nature Medicine* 27.5 (2021): 871-881.

Reviewer #3 (Remarks to the Author):

Overview of Analysis:

The authors take on the issue of determining whether putative AD subtypes observed in previous studies are actually intermediate stages in more generalized disease trajectories. They make the observation that previous cross-sectional neuroimaging papers are inherently unable to account for differences in disease trajectory, and thus use an longitudinal cohort of patients from 4 separate datasets in order to explore/cluster longitudinal changes in brain structure. They use patients w/ amyloid-confirmed AD from populations on 4 separate continents, and place all of them on a common timescale beginning at the time of AD diagnosis.

The authors first construct a multilevel model of normal age-related brain atrophy for each cohort. They accomplish this using amyloid-negative, cognitively unimpaired patients from the datasets in order to provide baseline atrophy levels to later compare changes observed in AD patients.

Each brain region (both cortical and subcortical, derived from the Desikan atlas) is modeled for atrophy over time using cohort _ subject as random effects, and age as the sole fixed effect.

The authors then apply a clustering model that groups patients by the atrophy dynamics measured in each brain region over time. Specifically, their approach treats each brain region as a separate response variable, with associated random slopes and intercepts. The intercepts represent the atrophy in a region at the time of diagnosis, whereas the slopes represent the speed of atrophy per region over the study period. In this case, atrophy is as measured by something called the w-score. This is essentially a Z-score re-adjusted for age and cohort.

The w-score metric is derived in relation to the normal degree of age-related atrophy that the authors estimate in their initial multilevel model of cognitively-unimpaired patients. The clustering model assumes a mixture of hierarchical models for w-scored atrophy. Random and fixed effects are fitted by Markov Chain Monte Carlo sampling, thus making the analysis Bayesian in nature (hence their title).

Disease duration (from time of AD diagnosis) was used as a cluster-specific random effect.

Intracranial volume and MRI field strength were used as population-wide fixed effects. By utilizing w-scoring and hierarchical modeling, the authors argue that they are sufficiently able to disentangle pathologic changes from normal aging. Furthermore, by

starting their modeling at the time of AD diagnosis, they argue that all patients are analyzed on a consistent timescale.

The authors test this mixture of hierarchical models using cluster sizes between 2 and 8. They ultimately state that 2- and 5-cluster solutions are superior solutions, as they minimize model deviation and featured the greatest degree of parameter convergence during MCMC sampling.

They ultimately decided to base further analyses on the 5-cluster model, it seems (they come up with names for these clusters later in the paper).

Finally, the authors turn their hierarchical model for AD atrophy into a classifier, and use this on a validation set of patients from each dataset. They population-average the model's predictions at the median disease duration, and then visualize the output by translating the predictions back to an "average" cortical surface per cluster. They then assess goodness of prediction by visually-comparing to an average cortical surface taken from the sum of observations at the median disease duration.

Of note, the authors also conducted 2 complementary analyses: They translated cluster-specific covariance matrices (ie covariance between random effect slopes and intercepts in each brain region) to graphs in which brain regions are nodes and pairwise covariances are edges. They used something called "nodal strength" in order to proxy the connectedness of atrophic changes each cluster, and then performed pairwise comparisons or nodal strength for different clusters.

They also performed multilevel modeling to compare differences in neurocognitive testing scores between clusters.

Findings/Implications:

-- The authors ultimately found that the 5 separate clusters they identified in their modeling had differing degrees of brain atrophy at the time of diagnosis, as well as the speed of atrophy observed across subsequent visits. However, by visual inspection of their model outputs, they ultimately concluded that all 5 clusters fell into one of two more general patterns of structural degeneration: mediotemporal vs. cortical spreading.

-- Various clusters seemed to converge at later time points in the disease course, based on the ultimate pattern of atrophy observed at the end-points of the study. Differing clusters had differing results on neurocognitive testing. Interestingly, patients with greater degrees of atrophy at time of diagnosis had greater premorbid intelligence, suggesting a protective effect from "cognitive reserve." Differing clusters had different degrees of nodal strength.

-- The ultimate point here (in my opinion): Though differing clusters were successfully derived from the datasets based on the dynamics of their longitudinal atrophy patterns, they ultimately did converge to two main pathways of structural change. This suggests that imaging subtypes previously observed on cross-sectional studies may be but intermediate stages in much more general patterns of structural change.

Strengths:

-- Generally, I believe that the complexity of the modelling appropriately matches the complexity of the problem. Mixed effects modelling with an overlying clustering approach seems appropriate to account for subject-level vs. population-level effects, while also accounting for the unresolved questions in the field regarding the existence of AD subtypes.

-- The data sources are very strong: 4 cohorts from 4 different continents, all with similar protocols for data acquisition and data harmonization.

-- I feel that the conclusions are made in the discussion are appropriate and quite feasible; furthermore, if this is indeed the first longitudinal imaging study of its kind, it is a very important contribution to the field.

Weaknesses:

Comment_14

-- The writing is somewhat inelegant in places. Though I realize this is an international group of researchers, I am wondering if a bit more could be done to smooth it in spots. The writing is superfluous in some spots, or does not go into sufficient detail in others.

Answer to the comment:

We want to thank the reviewer for pointing this out. We have tried to polish the text as much as possible. Further, we believe the comments from all three reviewers have helped filling the gaps where more details were needed or if something needed to be clarified or removed. Finally, we also sent the manuscript for English editing to be sure that the language is not affecting the text.

Comment_15

-- There is no mention of the patients' comorbidities and how these were/were not accounted for in the analysis. This is important, as the authors speculate that non-AD pathologies may play a role in the observed changes, but they never elaborate on it.

Answer to the comment:

We thank the reviewer for the comment. Unfortunately we have no biomarkers available for other types of pathologies such as Lewy bodies or TDP43. We only have a preprint available now regarding the prevalence of non-AD pathologies in the subtypes, in a small sample with antemortem MRI and postmortem neuropathological examination (Mohanty et al. 2021). Further, in our sample we only have white matter hyperintensities (WMH) for a very small number of patients and cannot add this information since it will not be representative of the total sample. We have however seen before that both location and prevalence of markers of small vessel disease are different across subtypes (Ferreira et al, 2018). This is briefly discussed in the manuscript on page 13. We can, however, only speculate in the present manuscript regarding these issues, and we have therefore now mentioned this as a limitation in the manuscript on page 15.

However, we really wanted to do something to address this issue, so we added a small table with additional information from the medical assessment of the AD patients that we found in the ADNI and JADNI samples (Table S9). However, this information is not

available for neither the AIBL nor the AddNeuroMed cohort and we do not want to bias our interpretations towards these cohorts. We have now referred to this table in the results section (see manuscript page 11, results, cluster characteristics).

Mohanty, Rosaleena, et al. "Susceptibility to postmortem (co)-pathologies in antemortem atrophy-based subtypes of Alzheimer's disease." *medRxiv* (2021).

<https://medrxiv.org/cgi/content/short/2021.09.06.21263162v1>

Ferreira, Daniel, et al. "The contribution of small vessel disease to subtypes of Alzheimer's disease: a study on cerebrospinal fluid and imaging biomarkers." *Neurobiology of aging* 70 (2018): 18-29.

Comment_16

-- The graph theory analysis needs more in-depth explanation. I am wondering if it is too much of a jump to suggest that stronger covariances from the modelling are automatically suggestive of structural correlation (at the very least, it is a claim that could use additional proof or analysis).

Answer to the comment:

We have tried to elaborate more on the reason behind the use of graph theory in our manuscript (see pages 9 and 10, complementary statistical analysis section). The main reasoning behind the graph theory utilization in this study was that covariance matrices can be summarized in fewer numbers, some of which are significant and can support inference in correlation between regional atrophy development or shared mechanisms in neurodegeneration. We did not want to suggest structural relation between brain regions just because of structural correlation and this is why we focused mainly on the longitudinal correlation between brain regions in the discussion (cluster slope covariance matrices) and not in cross sectional data correlation (cluster intercept covariance matrices).

Comment_17

-- I am also a bit confused what their implication is in comparing nodal strength between clusters. Perhaps that pathologic changes are more strongly propagated in certain of the clusters? I would be interested if they could back-correlate this with the trajectory of pathologic changes observed in some way.

Answer to the comment:

We agree completely with the reviewer on the utilization of nodal strength. Nodal strength was used as a summary of correlations in atrophy development since we focused more on atrophy slope covariances rather than atrophy intercepts. Regarding pathologic changes in AD, atrophy is what we assessed here. A recent study from our lab (Mohanty et al. 2021) focused on postmortem pathologies observed in the atrophy subtypes reported in the literature. In the updated manuscript (pages 14 and 15) we made an effort to comment on the potential differences between the mediotemporal and cortical atrophy pathways in terms of glucose metabolism and amyloid accumulation in the context of the default mode network. The two pathways have also consistently different regional nodal strength properties. Although the speculation that we made in our discussion is early and maybe farfetched, future studies that will include amyloid, glucose, atrophy, and tau longitudinal data for the different AD subtypes will assess those questions.

Mohanty, Rosaleena, et al. "Susceptibility to postmortem (co)-pathologies in antemortem atrophy-based subtypes of Alzheimer's disease." *medRxiv* (2021).

<https://medrxiv.org/cgi/content/short/2021.09.06.21263162v1>

Comment_18

-- It is not entirely clear to me why the 5-cluster solution was ultimately chosen over the 2-cluster solution. The authors tout them as being equally good, but ultimately elect to discuss the 5-cluster solution for the majority of the paper.

Answer to the comment:

We thank the reviewer for pointing this out. While this was discussed in our previous methodological publication (Poulakis et al. 2020) as well as in another publication considering advanced clustering techniques for the assessment of neurological disorders (Yang et al. 2021), we failed to elaborate it in our current manuscript. In the aforementioned applications the 2-cluster solution normally corresponds to a solution that separated the sample to a severe and an unaffected or less affected group of individuals/patients. This happens because severity as a dimension of the disease is always more prominent than typicality (referring to existence of atypical disease subtypes). This causes clustering techniques to find a 2-cluster solution as the most optimal one, rather than one with more clusters that includes some interesting cluster formations from an exploratory perspective. This is now clarified on page 9 of the manuscript.

Poulakis, Konstantinos, et al. "Fully Bayesian longitudinal unsupervised learning for the assessment and visualization of AD heterogeneity and progression." *Aging (Albany NY)* 12.13 (2020): 12622.

Yang, Zhijian, et al. "Disentangling brain heterogeneity via semi-supervised deep-learning and MRI: dimensional representations of Alzheimer's Disease." *arXiv preprint arXiv:2102.12582* (2021).

Comment_19

-- Do the authors provide a quantitative basis for their claim that the 5 clusters ultimately boil down into cortical vs. mediotemporal spreading categories? Or is it just visual?

Answer to the comment:

The summary of the 5 clusters in two paths of neurodegeneration is partially a qualitative result of the clustering model interpretation but is also based on quantitative data. From the mean atrophy trajectories as well as the covariance similarities of the five clusters, one can conclude that with a lag of 1-3 years of disease progression the atrophy levels and patterns of the minimal atrophy, limbic predominant and limbic predominant + clusters of patients are very similar compared to the hippocampal sparing pattern. For the diffuse atrophy pattern is challenging to predict which atrophy pathway it follows. More longitudinal data before the AD dementia onset is needed to assess this, we can only speculate (see page 14). Observing carefully the existence of only two main atrophy trends in the dataset we named the two atrophy pathways as mediotemporal and cortical.

#End of reviewers' comments

Reviewers' Comments:

Reviewer #1:

Remarks to the Author:

Thank you for your responses. However, there are still points unclear:

For the validation part: The current approach still does not try to cluster in independent data and compare the subject characteristics, which remains a weak aspect of this research. There will always be a cluster solution, the question is how robust is the solution (or, does the solution depend on your data). Given the large data set, there are many different ways to address this, my suggestion of repeating clustering within the studies and assess stability of cluster membership was only one of many.

For the minimal atrophy subgroup, which was the largest, the response remains unconvincing that these individuals indeed don't show atrophy. It would be informative to show a statical map (e.g., t-stats) thresholded on significance in order to assess the atrophy patterns of subtypes compared to controls. Possibly per 'stage'. As it is presented now, it is a bit misleading/suggestive that there is no atrophy at all in these dementia individuals.

For the cluster probability assignment supplementary figure s6: Please indicate how uncertainty was determined (in other words, why only 5 individuals uncertain mapping, when there are more subject points in the middle of the figure). This figure further suggest that a large group of individuals the minimal atrophy cluster and the limbic predominant atrophy cluster subjects fall in between both, is this related to the amount of atrophy/disease stage?

Reviewer #2:

Remarks to the Author:

Whilst I do think this work has the potential to be an interesting and important study, I was not convinced by the responses to reviewers from the authors. Even after revision the validation is very weak, the claims are overstated in places and the methodological justification is poor. Specific comments below.

Major comments

The biggest issue with the study is the lack of validation. In their response the authors stated that their current validation serves two purposes. The first is as a sanity check that new individuals assigned to the clusters look like the clusters they are assigned to, which would be expected to be the case for most clustering techniques regardless of the quality of the clustering. The second is to allow them to perform statistical analysis, which is not to do with validation, but with understanding statistical differences between clusters. In my opinion it is misleading to refer to either of these analyses as validation. A validation would demonstrate that the clusters they have identified are stable in an external dataset. There are two possible ways they could perform this validation:

- The clustering could be re-run separately in ADNI and (J-ADNI + AIBL).
- Given that the amyloid negative and positive AD patients had the same cluster frequency, all individuals with AD regardless of amyloid status could be included to create two datasets – one from ADNI and one from (J-ADNI + AIBL + AddNeuroMed) – and the clustering could be run separately in each.

Re-running the clustering using X-fold cross-validation in the original discovery dataset is another option, which is weaker than external validation but would be better than no validation.

Another major issue is that the results do not support the claims made by the authors. The central premise of the manuscript is to assess the "subtype vs. stage" hypothesis. This is a big claim that the current study design doesn't address convincingly. The authors model clusters that are on a common

timeline from AD onset, however it is well known that at AD onset individuals are at a range of different underlying pathological stages along the disease and so each of the subtypes they find can represent a different portion of the overall disease trajectory. They try to address this by visually piecing together the trajectories and mapping them onto two common trajectories. This ultimately amounts to the same evidence others have given in clustering studies, where they identify cross-sectional clusters and then speculate post-hoc which may be earlier or later stages of each. The inclusion of short-term longitudinal data does not significantly enhance their claim as the longitudinal portions are short – less than two years in most cases and so can be considered as closer to cross-sectional with respect to the full disease time course. Additionally, the pre-symptomatic portion of the trajectory is not modelled and so they cannot provide insight on the early stages of atrophy. The dataset is also small due to the requirement for longitudinal follow-up, with only 320 individuals with AD being used to estimate the clusters, meaning it is possible the ability to separate subtle cluster differences is underpowered compared to the larger cross-sectional datasets that have been used by others. The value of the study is in performing a longitudinal subtyping study on an easily interpretable timescale (time from AD onset). This is an important contribution, and in my opinion the manuscript would be much stronger if it was rewritten to centre around this premise rather than making overstated claims.

Finally, there are a number of methodological issues that have not been addressed.

- I remain sceptical about the choice of the number of clusters. The authors have not provided a reference for using the percentage of MCMCs with higher autocorrelation as a measure of clustering quality besides their own paper, which I looked through and couldn't find a reference in either. This needs proper justification. The model deviance presented in the supplementary material suggests that 4 clusters is the best solution rather than 2 or 5.
- As far as I can tell the ad-hoc fitting method of estimating linear slopes and then adding quadratic terms provides no guarantee of achieving a global optimum and so needs to be re-run from several starting points to demonstrate that convergence to the same solution is achieved from different points within the space.
- The condensing of five pathways into two can and should be supported by quantitative analysis. There is no statistical comparison problem if the authors simply compare the estimated model parameters at each time point across each of the five clusters to see how well the modelled brain maps match across different points of the trajectories across clusters.

Minor comments

Table S1 is useful in understanding the study data, however it doesn't show where these individuals fall in terms of time from AD onset. It would be beneficial to show a histogram or similar of time from AD onset vs. number of subjects to gain an idea of which portions of the trajectories are well sampled. It would be useful to also show this information after assigning individuals into clusters to check that the trajectories are well sampled in each cluster.

In the methods the authors refer to atrophy levels, when I think they mean volumes. Atrophy implies a change in volume measured over time, but I think in this case cross-sectional volumes were input into a longitudinal model to assess atrophy over time and so the inputs are volumes, not atrophy levels.

This sentence needs revising: "The word complex is used to showcase that probabilistic mixture modelling (number of clusters) is combined with mixed effect modelling (number of brain regions) in one model definition."

The longitudinal clustering model would benefit from being specified mathematically.

Why are field strength and eTIV included in the longitudinal clustering model, but age and cohort effects regressed out in the CU population? Why not do both at once?

The exact calculation for the “percentage of MCMCs with higher autocorrelation compared to the majority of MCMCs” needs writing down.

The claim in the introduction that “the most urgent questions are ... if these subtypes finally converge at advanced stages of the disease” needs referencing given that the authors agree that it is inevitable. Otherwise this should be removed as a key insight of the study.

Reviewer #3:
Remarks to the Author:
My comments below:

Comment 14:

· I feel that the authors have responded appropriately. I appreciate their willingness to address areas where changes in writing style would strengthen the overall manuscript.

Comment 15:

· The reference to Ferreira et al. appears to account for potential differences in comorbidities between AD subtypes. The authors’ acknowledgement of limited data with respect to comorbidities is reasonable and, in my opinion, the inclusion of an additional citation to this point adequately respects an area of study that is yet unexplored. The newly-added supplementary table S9 is also quite comprehensive in terms of the body systems that it addresses in the ADNI and J-ADNI cohorts. Given the paper’s focus on neuroimaging, I do not believe that it is necessary at this time to pursue further in-depth analysis of non-neurologic disease burden.

Comment 16:

· I appreciate the authors’ elaboration of their use of graph theoretical approaches. In looking at the manuscript, I am satisfied by their statement that “nodal strength calculation was not used as the main analytical step in this study but only to help summarize the information from the cluster covariance matrices.”

· I do think that the authors would do well to insert several sentences further delineating the difference between cross sectional atrophy correlations and longitudinal correlations. They do this well in their response document, wherein they state: “We did not want to suggest structural relation between brain regions just because of structural correlation, and this is why we focused mainly on the longitudinal correlation between brain regions”

· Given the importance of the time axis in the study’s main conclusions, I feel that very clearly stating the essential differences in longitudinal vs. cross-sectional structural correlations will improve the Discussion section.

Comment 17:

· I believe that the authors’ explanation of their use of nodal strength clarifies this issue, and the addition of additional material to pages 14 and 15 is appropriate. I feel that they have made sufficient adjustments to comment on pathological changes in the respective subtypes, and the role of nodal strength is clearer now. My only concern is that reviewers have cited a preprint on medRxiv here (which is not peer-reviewed).

Comment 18:

· The authors in their response provide a very intuitive and well-backed rationale for electing the 5-cluster solution over the 2-cluster solution. Based on their explanation, the 5-cluster solution provides a greater degree of resolution by which to probe structural changes in the brain whereas the 2-cluster solution stratifies severe vs. mild changes on a much coarser level. I do like this explanation quite a bit, however, I feel that it should be reflected in their manuscript. As it currently stands, they write

"Since different spatial atrophy subtypes are more interesting from an exploratory perspective, we chose to interpret the results of the 5-cluster solution." I feel that this is insufficient relative to what they have provided in the response, which is clearer and has greater backing in evidence.

· I also find slight issue with the use of "interesting", as it could be seen as assuming an opinion on the part of a reader.

All changes in the manuscript are marked in yellow.

Reviewers' comments:

Reviewer #1 (Remarks to the Author):

Comment 1

Thank you for your responses. However, there are still points unclear:

For the validation part: The current approach still does not try to cluster in independent data and compare the subject characteristics, which remains a weak aspect of this research. There will always be a cluster solution, the question is how robust is the solution (or, does the solution depend on your data). Given the large data set, there are many different ways to address this, my suggestion of repeating clustering within the studies and assess stability of cluster membership was only one of many.

Answer to comment 1

We agree with the reviewer that validation is very important. Therefore, we have now repeated the clustering in the ADNI cohort's longitudinal data and the JADNI+AIBL cohorts separately (**see page 8**). The reason why we merged JADNI and AIBL lies on the small AIBL sample with repeated measurements (it was also a suggestion of the second reviewer). The separate datasets showed similar results to the combined dataset (**see page 11 of the manuscript, figures S7 and S8 in the supplementary material**). The low prevalence of some patterns such as diffuse atrophy and hippocampal sparing (**see table 2**) made them be discovered as outliers in the separate datasets' analyses (**see supplementary table S6**). This normally happens in clustering, when the split of a population divides an already small group, such as a subtype of low prevalence (which we already know is the case of the hippocampal sparing subtype from the AD subtypes literature, see Ferreira and colleagues, 2020), into small segments. This reduces the chances that the algorithm will discover the complete cluster (all cases), since all cases are not completely similar. The ones closer to another more prevalent cluster are attached to it, and the most distant cases become an outlier cluster (**see pages 11 and 16-17 for results and discussion**). The observation proximity idea is well depicted in **supplementary figure S5 too**.

To better explain our validation approach which we believe provide important information, we made a simulated example of how clustering and evaluation in new data based on a supervised model can fail, if a similar evaluation method to supervised analysis is used. We clustered data that are obviously clustered in only two clusters. Then we trained a logistic regression to the training data and the clustering allocations as labels (in the same fashion as our manuscript). We then tested the logistic regression in new data that come from the same and completely different populations as the training data. In the case of similar populations, the classification is accurate. In the case where the new and unseen data came from different populations, the classifier (logistic regression) returned probabilities that clustered the data in two classes, but it was obvious that the model failed to "understand that the classification is wrong". By taking the difference between each new data point and the mean of each of the original two classes, it is possible to identify that the observations of one of the classes do not match the class where the observations are classified into. In the same fashion, we calculated differences between new data cluster means and the fitted values of each cluster of our trained clustering model. The difference showcases this problem and helps us understand the generalization of a clustering models' result to new data.

See simulation and code for it at the end of this document (page 12 of the answer to the reviewers document).

To summarize, the original data that were used in the simulated example included sampling from two distributions with means of 50 and 75 years of age. The completely new simulated data have means of 10 and 80 years of age.

- If we subtract from the 1st class mean (50 years of age) the mean of the completely new data (test data) that were classified as class 1 (10 years of age), we find that the test data have a difference of 40 years in average from the class where they were classified, which is very large.
- If we subtract from the 2nd class mean (75 years of age) the mean of the completely new data (test data) that were classified as class 1 (80 years of age), we find that the test data have a difference of 5 years in average from the class where they were classified, which is not very large.

This approach is what we applied in our study. **Figure 3** of the manuscript shows differences in means from each original cluster to its equivalent predicted cluster at a brain region of interest level. The patterns look similar for each of the 5 clusters which means that the supervised predictions are based of sensible classifications of patients into patterns of atrophy based on previous clustering. The description of the validation method is on **page 8**, the results of the validation method are presented on **page 11-12** and they are discussed on **page 16-17**.

Ferreira, Daniel, Agneta Nordberg, and Eric Westman. "Biological subtypes of Alzheimer disease: A systematic review and meta-analysis." *Neurology* 94.10 (2020): 436-448.

Comment 2

For the minimal atrophy subgroup, which was the largest, the response remains unconvincing that these individuals indeed don't show atrophy. It would be informative to show a statical map (e.g., t-stats) thresholded on significance in order to assess the atrophy patterns of subtypes compared to controls. Possibly per 'stage'. As it is presented now, it is a bit misleading/suggestive that there is no atrophy at all in these dementia individuals.

Answer to comment 2

We understand the reviewers concern and we have now done extensive complementary work in our data and we can show the reasons why the largest group (minimal atrophy) show very little atrophy at the simulated AD onset age compared to the cognitively unimpaired sample.

We use the time at the AD onset from the clinical records. Many patients got the clinical AD diagnosis many years before the first MRI. Our modeling reveals patterns of atrophy after the clinical AD onset controlled for variability in brain morphology and brain changes occurring in healthy aging. As it is seen in **Figure 1** of the manuscript (longitudinal grey matter patterns in the cognitively unimpaired population (**Figure 1A**), longitudinal grey matter patterns in the AD population (**Figure 1B**), comparison between AD and the cognitively unimpaired population at different ages at population level (**Figure 1C**)), differences are not very high until the patients reach ages above 75 years of age (see row A for cognitively unimpaired and row B for AD, colors are orange corresponding to values around 0) (**results, page 10**). Atrophy is observed in most of the AD patients at the age 75 (which is also the age of most of our AD dataset at the first visit) but is not profound at the estimated AD onset. However, after some experimentation on the cutoff for imaging threshold in the fitted values of the model output of the manuscript, we see as expected that when we reduce the imaging cut off for the AD clusters' fitted values, from 1.6 (see **figure 2** of the manuscript) to 0.5 standard deviations (see **supplementary figure S4**) below the cognitively unimpaired population (20% below the mean of the cognitively unimpaired population), atrophy in the mediotemporal lobe is prominent at the earliest stage of dementia (AD onset) (**page 10**). This finding is in line with the literature in supervised MRI studies on AD patients against cognitively unimpaired individuals, where the mediotemporal atrophy is one of the first discriminants of the two diagnostic labels (Echávarri et al., 2011; Raji et al., 2009). We decided to use in the manuscript two different figures in the same manner as the key paper of Whitwell and colleagues did (Whitwell et al 2012). One figure has a cut off at 95% (1.6 standard deviations below cognitively unimpaired mean value) of the cognitively unimpaired population's atrophy values (assuming atrophy normality distributions in the cognitively unimpaired MRI data) and is added in the main manuscript on **pages 10 and 11 (Figure 2)**. **As supplementary information we added the same data but with a cut off value at 20% (0.5 standard deviations below cognitively unimpaired mean value) of the cognitively unimpaired population's atrophy values (see supplementary figure S4)**. The first set of images can be seen as a corrected (more conservative) version of the second set of images. The threshold of 1.6 standard deviations is commonly used in the literature and discussed extensively by Jack and colleagues (Jack et al., 2017). As for the lower threshold (0.5 standard deviations below the controls), it is used as an uncorrected version of the first threshold. **This is clarified and discussed in the manuscript on page 7-8 and page 13.**

Echávarri, Carmen, et al. "Atrophy in the parahippocampal gyrus as an early biomarker of Alzheimer's disease." *Brain Structure and Function* 215.3-4 (2011): 265-271.

Raji, Cyrius A., et al. "Age, Alzheimer disease, and brain structure." *Neurology* 73.22 (2009): 1899-1905.

Jack Jr, Clifford R., et al. "Defining imaging biomarker cut points for brain aging and Alzheimer's disease." *Alzheimer's & Dementia* 13.3 (2017): 205-216.

Comment 3

For the cluster probability assignment supplementary figure s6: Please indicate how uncertainty was determined (in other words, why only 5 individuals uncertain mapping, when there are more subject points in the middle of the figure). This figure further suggest that a large group of individuals the minimal atrophy cluster and the limbic predominant atrophy cluster subjects fall in between both, is this related to the amount of atrophy/disease stage?

Answer to comment 3

In order to assess uncertainty, we assessed the table with the probability of a patient to belong in any of the five clusters (cluster/patient probability matrix). We followed two paths. Firstly, we followed the established rule (see clustering procedure, section 3, in Komarek and colleagues, 2013 and Komarek and colleagues 2014) that an observation is uncertainly classified if none on the cluster probabilities of that specific observation is higher than 50%. Practically, this means that if one patient fails to be classified with at least 0.5 probability to one of the five clusters then the patient is uncertainly classified under our implementation.

Secondly, we analyzed the variance of this table that has a dimension of 5 columns and 571 rows, by means of principal components. In that way we managed to summarize the probabilities of patients to belong in any of the five clusters to less than five columns (one for each cluster) (see Poulakis et al, 2020, 2021). Only four out of the five hundred and seventy-one analyzed components were above near zero values (see the eigen values figure above).

In figure s6 you can see a 3d plot of the first three components which summarize most of the variance of the cluster/patient probability matrix. As the reviewer correctly points out, some subjects are in the middle of the figure. However, if this figure could be seen in a 3-dimensional space one would have noticed that the points that look like they are in the middle are not really in the middle but closer to the corners. Below we attach one more figure (the figure below with PC1-4 diagonal scatterplot matrix) to support this claim. This figure shows the relationship between the four principal components, but in pairs of components. The colors represent the same clusters as in figure s6. Although some conclusions about the patients' assignment into clusters can be drawn from the 3d plot of figure s6, much information can be found in the pairwise component plot too. For instance, the first component shows a good separation between minimal atrophy (purple dots) and limbic predominant (blue dots), while the second component shows a good separation between two groups of clusters: minimal atrophy (purple dots)/limbic predominant (blue dots) versus limbic predominant atrophy plus (red dots)/diffuse atrophy(green)/hippocampal sparing(cyan). The third principal component summarizes the information of the separation between diffuse atrophy (green) and limbic predominant plus (red). Finally, the fourth component focuses on the separation between hippocampal sparing and all the other patients in the validation dataset. The cluster/patient probability matrix is visualized through principal components (covariance matrix visualization) that include as much information of the data variance as possible. However, the complete information about the data is not depicted with the figure s6 since dimensionality reduction was used for the sake of visual evaluation of the clustering result.

It is correct that some patients of the minimal and limbic predominant atrophy clusters have probabilities that are close to their borders of the clusters. This is clearly depicted in their atrophy severity values (Figure 2). This is the very point that we have been trying to make throughout the manuscript (see **page 13-14 for discussion on the similarities between MA and LPA clusters**). Patients of the minimal atrophy cluster differ from the ones of the limbic predominant cluster mainly in terms of atrophy severity at the baseline, and they begin to show similar atrophy values to the limbic predominant cluster after approximately two years. The reviewer correctly points out that this proximity in the minimal atrophy and limbic predominant patients' probabilities are related to the atrophy disease stage. We have also complemented the legend of the **figure S5** with additional information to make the clustering procedure of the

validation dataset clearer.

Komárek, Arnost, and Lenka Komárková. "Clustering for multivariate continuous and discrete longitudinal data." *The Annals of Applied Statistics* (2013): 177-200.

Komárek, Arnošt, and Lenka Komárková. "Capabilities of R package mixAK for clustering based on multivariate continuous and discrete longitudinal data." *Journal of Statistical Software* 59.1 (2014): 1-38.

Poulakis, Konstantinos, et al. "Longitudinal deterioration of white-matter integrity: heterogeneity in the ageing population." *Brain communications* 3.1 (2021): fcaa238.

Poulakis, Konstantinos, et al. "Fully Bayesian longitudinal unsupervised learning for the assessment and visualization of AD heterogeneity and progression." *Aging (Albany NY)* 12.13 (2020): 12622.

Reviewer #2 (Remarks to the Author):

Whilst I do think this work has the potential to be an interesting and important study, I was not convinced by the responses to reviewers from the authors. Even after revision the validation is very weak, the claims are overstated in places and the methodological justification is poor. Specific comments below.

Major comments

Comment 4

The biggest issue with the study is the lack of validation. In their response the authors stated that their current validation serves two purposes. The first is as a sanity check that new individuals assigned to the clusters look like the clusters they are assigned to, which would be expected to be the case for most clustering techniques regardless of the quality of the clustering. The second is to allow them to perform statistical analysis, which is not to do with validation, but with understanding statistical differences between clusters. In my opinion it is misleading to refer to either of these analyses as validation. A validation would demonstrate that the clusters they have identified are stable in an external dataset. There are two possible ways they could perform this validation:

- The clustering could be re-run separately in ADNI and (J-ADNI + AIBL).

- Given that the amyloid negative and positive AD patients had the same cluster frequency, all individuals with AD regardless of amyloid status could be included to create two datasets – one from ADNI and one from (J-ADNI + AIBL + AddNeuroMed) – and the clustering could be run separately in each. Re-running the clustering using X-fold cross-validation in the original discovery dataset is another option, which is weaker than external validation but would be better than no validation.

Answer to comment 4

We agree with the reviewer that validation is very important. Therefore, we have now repeated the clustering in the ADNI cohort's longitudinal data and the JADNI+AIBL cohorts separately (see page 8). The reason why we merged JADNI and AIBL lies on the small AIBL sample with repeated measurements.

The separate datasets showed similar results to the combined dataset (see page 11 of the manuscript, figures S7 and S8 in the supplementary material). The low prevalence of some patterns such as diffuse atrophy and hippocampal sparing (see table 2) made them be discovered as outliers in the separate dataset analyses (see supplementary table S6). This normally happens in clustering, when the split of a population divides an already small group, such as a subtype of low prevalence (which we already know is the case of the hippocampal sparing subtype from the AD subtypes literature, see Ferreira and colleagues, 2020), into small segments. This reduces the chances that the algorithm will discover the complete cluster (all cases), since all cases are not completely similar. The ones closer to another more prevalent cluster are attached to it, and the most distant cases become an outlier cluster (see pages 11-12 and 17 for results and discussion). The observation proximity idea is well depicted in supplementary figure S5 too.

To better explain our validation approach which we believe provide important information, we made a simulated example of how clustering and evaluation in new data based on a supervised model can fail, if a similar evaluation method to supervised analysis is used. We clustered data that are obviously clustered in only two clusters. Then we trained a logistic regression to the training data and the clustering allocations as labels (in the same fashion as our manuscript). We then tested the logistic regression in new data that come from the same and completely different populations as the training data. In the case of similar populations, the classification is accurate. In the case where the new and unseen data came from different populations, the classifier (logistic regression) returned probabilities that clustered the data in two classes, but it was obvious that the model failed to “understand that the classification is wrong”. By taking the difference between each new data point and the mean of each of the original two classes it is possible to identify that the observations of one of the classes do not match the class where the observations are classified into. In the same fashion, we calculated differences between new data cluster means and the fitted values of each cluster of our trained clustering model. The difference showcases this problem and helps us understand the generalization of a clustering models' result to new data.

See simulation and code for it at the end of this document (page 12 of the answer to the reviewers document).

To summarize, the original data that were used in the simulated example included sampling from two distributions with means of 50 and 75 years of age. The completely new simulated data have means of 10 and 80 years of age.

- If we subtract from the 1st class mean (50 years of age) the mean of the completely new data (test data) that were classified as class 1 (10 years of age), we find that the test data have a difference of 40 years in average from the class where they were classified, which is very large.
- If we subtract from the 2nd class mean (75 years of age) the mean of the completely new data (test data) that were classified as class 1 (80 years of age), we find that the test data have a difference of 5 years in average from the class where they were classified, which is not very large.

This approach is what we applied in our study. Figure 3 of the manuscript shows differences in means from each original cluster to its equivalent predicted cluster at a brain region of interest level. The patterns look similar for each of the 5 clusters which means that the supervised predictions are based of sensible classifications of patients into patterns of atrophy based on previous clustering. The description of the validation method is on page 8, the results of the validation method are presented on page 11-12 and they are discussed on page 16.

Ferreira, Daniel, Agneta Nordberg, and Eric Westman. "Biological subtypes of Alzheimer disease: A systematic review and meta-analysis." *Neurology* 94.10 (2020): 436-448.

Comment 5

Another major issue is that the results do not support the claims made by the authors. The central premise of the manuscript is to assess the “subtype vs. stage” hypothesis. This is a big claim that the current study design doesn't address convincingly. The authors model clusters that are on a common timeline from AD onset, however it is well

known that at AD onset individuals are at a range of different underlying pathological stages along the disease and so each of the subtypes they find can represent a different portion of the overall disease trajectory. They try to address this by visually piecing together the trajectories and mapping them onto two common trajectories. This ultimately amounts to the same evidence others have given in clustering studies, where they identify cross-sectional clusters and then speculate post-hoc which may be earlier or later stages of each. The inclusion of short-term longitudinal data does not significantly enhance their claim as the longitudinal portions are short – less than two years in most cases and so can be considered as closer to cross-sectional with respect to the full disease time course. Additionally, the pre-symptomatic portion of the trajectory is not modelled and so they cannot provide insight on the early stages of atrophy. The dataset is also small due to the requirement for longitudinal follow-up, with only 320 individuals with AD being used to estimate the clusters, meaning it is possible the ability to separate subtle cluster differences is underpowered compared to the larger cross-sectional datasets that have been used by others. The value of the study is in performing a longitudinal subtyping study on an easily interpretable timescale (time from AD onset). This is an important contribution, and in my opinion the manuscript would be much stronger if it was rewritten to centre around this premise rather than making overstated claims.

Answer to comment 5

We agree with the reviewer that the lack of data for each patient during the complete disease trajectory is a limiting factor with respect to the interpretability of our results regarding the staging problem in AD subtyping. However, we do not believe that 320 amyloid positive patients with repeated measurements MRI are a small dataset, since there is no other study to date addressing this question with larger numbers, and our analysis includes data from different continents to increase sample variability. Moreover, our main validation (based on the validation dataset) is not based on a test set, but we actively investigated whether AD patients with different atrophy patterns exist in the validation dataset (please see the answer to comment 4 for a deeper explanation of our main validation approach).

Regarding the value of the study, in our analysis we combined a correction of the MRI's for age-related normal brain morphological changes, which has not been done in other studies before (longitudinal grey matter patterns in the cognitively unimpaired population (**Figure 1A**), longitudinal grey matter patterns in the AD population (**Figure 1B**), comparison between AD and the cognitively unimpaired population at different ages at population level (**Figure 1C**) with longitudinal clustering (see **pages 10-11, and 17**). An advantage of this approach over other approaches, is that the correction of age as an environmental factor in our design has reduced the heterogeneity of our data related to aging, which is an issue when assessing AD subtypes (see Feczko and colleagues (2019) for a technical review on the limitations of heterogeneity assessment, section "Human Population Is Profoundly Heterogeneous Across Multiple Dimensions"). Regarding the short MRI follow ups, we believe that a strong methodological aspect of this study compared to the literature, is the reconstruction of longitudinal subtype-atrophy profiles over the dementia part of the AD continuum, based only on longitudinal individual patients' data that comprised short segments of the disease continuum. This is now discussed on **page 17**. Moreover, in our opinion, cross-sectional studies by default cannot be compared to any longitudinal study, because the lack of individual trajectories even with short follow up times, does not allow for within subject variability assessment (random effect at a subject level). We strongly believe that we do not over interpret our results. Our conclusions are not solely based in the patterns of atrophy discovered in the discovery dataset, but in a large set of gathered evidence around the data at question. 1) The patterns of atrophy, 2) the demographical characteristics of the patients that show each pattern of atrophy, 3) the covariance structure of each cluster's longitudinal patterns of atrophy, 4) the validation of the results both with a validation dataset and separate clustering analyses in the cohorts, 5) the conclusions based on groups of clusters that we later called pathways of neurodegeneration are some of the results that yielded our conclusions. Moreover, we reserved ourselves from interpreting all clusters of AD patients with the same certainty. For example, we do not claim that the diffuse atrophy pattern is a subtype of AD because our data are not enough to support this claim (see **page 15, second paragraph**). Instead, we acknowledge in the limitation section that with data that span all the AD continuum we will understand the phenomenon that we investigated in our study even more completely (more discussion in the limitations of the study is now added on **page 17**).

Feczko, Eric, et al. "The heterogeneity problem: Approaches to identify psychiatric subtypes." *Trends in Cognitive Sciences* 23.7 (2019): 584-601.

Comment 6

Finally, there are a number of methodological issues that have not been addressed.

- I remain sceptical about the choice of the number of clusters. The authors have not provided a reference for using the percentage of MCMCs with higher autocorrelation as a measure of clustering quality besides their own paper, which I looked through and couldn't find a reference in either. This needs proper justification. The model deviance

presented in the supplementary material suggests that 4 clusters is the best solution rather than 2 or 5.

Answer to comment 6

We appreciate the comment and understand the concern. We have used this method of selection of clusters in previous papers (Poulakis et al., 2020; Poulakis et al., 2021). In the previous round of review we explained some aspects that we expand here now, including reasoning behind this approach and our decision. The optimization of very large Bayesian models has always “blind” spots that needs quality analysis that exceeds the likelihood assessment (Barbos et al., 2018). We completely agree with the reviewer that further explanation and writing of the formula of assessment is needed and **have added it under the supplementary table S3**. We have explained better our approach and **added one more reference (methods, page 7)**. Moreover, at **page 6, we have added 11 references** regarding the use of the longitudinal clustering model in different application. Readers can now refer to the literature to better understand the model definition and implementation if they want to. We have also explained why we selected the five and not two or four cluster solution (results, **page 10**). We also agree with the reviewer that the solution with four clusters has a lower deviance than the five-cluster solution. However, the number of means with high autocorrelation is numerically higher for the four-cluster solution. Another feature of the variable selection is the visual assessment of the results. The four-cluster solution compresses two clusters of the five-cluster solution to one which unfortunately produced a cluster with high within cluster variance (data not shown in the manuscript). The LPA and MA longitudinal patterns of atrophy look very similar in **Figure 2** of the manuscript. However, when an uncorrected version of that figure is added to the interpretation (**supplementary figure S4**), then we observe that there are considerable intercept differences between LPA and MA that are better captured by the five-cluster solution. Importantly, assessment of longitudinal clustering models is a very demanding task, since no conventional distance quality indices (such as CH, Silhouette or Dunn indices, or Hopkins statistic) can be employed in the absence of a distance matrix summarizing the dataset. More research in this direction should be done now that longitudinal datasets are becoming increasingly available. We would like to take the opportunity here to explain why there is not a lot of material in Bayesian clustering assessment in the literature currently. In this study we used a mixture model for the clustering. The model was optimized with a Bayesian optimizer and the quality of MCMC is assessed in Bayesian models with some specific tools independently of the supervised/unsupervised nature of it. Thus, it can be very hard to find scientific literature under the keywords Bayesian model cluster assessment, but not so hard to find literature in Bayesian model quality assessment. We cite some research articles/reviews about assessment of MCMC output so that it's made clearer why we assessed MCMC with autocorrelation measures (Roy Vivekanda (2020) for autocorrelation discussion; Vats and colleagues (2019) for discussion in the importance of smart starting values, thinning that we also applied, many parallel chains that we also applied, and autocorrelation; Toft and colleagues (2007) for autocorrelation assessment) (see **pages 5-8, 10, 17**).

Poulakis, Konstantinos, et al. "Longitudinal deterioration of white-matter integrity: heterogeneity in the ageing population." *Brain communications* 3.1 (2021): fcaa238.

Poulakis, Konstantinos, et al. "Fully Bayesian longitudinal unsupervised learning for the assessment and visualization of AD heterogeneity and progression." *Aging (Albany NY)* 12.13 (2020): 12622.

Roy, Vivekananda. "Convergence diagnostics for markov chain monte carlo." *Annual Review of Statistics and Its Application* 7 (2020): 387-412.

Vats, Dootika, et al. "Analyzing MCMC output." arXiv preprint arXiv:1907.11680 (2019).

Toft, Nils, et al. "Assessing the convergence of Markov Chain Monte Carlo methods: an example from evaluation of diagnostic tests in absence of a gold standard." *Preventive veterinary medicine* 79.2-4 (2007): 244-256.

Bărbos, A. C., et al. "Clone MCMC: parallel high-dimensional Gaussian Gibbs sampling." (2018).

Comment 7

- As far as I can tell the ad-hoc fitting method of estimating linear slopes and then adding quadratic terms provides no guarantee of achieving a global optimum and so needs to be re-run from several starting points to demonstrate that convergence to the same solution is achieved from different points within the space.

Answer to comment 7

We apologize for not explaining our method more clearly in the manuscript. To make our approach clear to the reviewer we would like to stress that we simulated 500000 samples of the Gibbs sampler for the quadratic model. We did not bound the model parameters that were already optimized in the linear model in any way. When the quadratic model optimization took place, all the model parameters were re-optimized as if optimization had just started with the original data. Only initial values for the parameters were additionally provided there. This means that no ad hoc addition was implemented in the results of the clustering. The full quadratic model was optimized independently of the linear model and reached optimum values. The quadratic model was optimized from 28 different initialization points summing up to 14000000 parameter samples to do the final model selection.

If we initialize the quadratic model's intercepts and slopes with the maximum a-posteriori estimates of all linear models (all the candidate models, different initializations with different initial points and number of clusters) we provide the algorithm with the best guess that can be given in terms of intercept and slope estimates. In the same manner Lynch (2007), advice on giving maximum likelihood estimates as best initial values for an MCMC if they exist. An implemented version of this principle is provided by the SAS software's user guide on MCMC method specification (https://support.sas.com/documentation/cdl/en/statug/68162/HTML/default/viewer.htm#statug_mi_details32.htm) and by a model-based clustering approach for high dimensional data (Berge et al., 2012), among other implementations. Similarly, in our implementation we start with initial values for the means of the clusters (multivariate normal distribution means vector) based on the literature (Vats et al., 2019; Haario et al., 2005; Gelfand, 2000, Barbos et al., 2018). The optimizer will find optimal values for a number of parameters in the linear model that includes an intercept and slope (linear term) only. Later on and when the first part of the optimization is finished the model is redefined as quadratic.

When the model is turned into quadratic by adding the "third", quadratic term in the model definition, the optimization scheme must estimate three types of parameters: intercepts, a linear and a quadratic term. The quadratic model is, by definition, less efficient in optimization terms (reach stationary MCMC distributions) than a linear model since it is overparameterized (Haario et al., 2005; Gelfand, 2000). The model optimizer will:

- 1) not alter the intercept a lot because it is already fitted well by the linear model based on the likelihood of the data and the weakly informative priors (Richardson and Green 1997).
- 2) slightly alter the linear term estimations based on the new data fit (quadratic fit) to facilitate the new model definition (linear plus quadratic term).
- 3) find optimal values for the quadratic terms (other than zero, if and only if there is a quadratic pattern in the data).

On the other hand, when the model is fitted as quadratic with no "smart" initialization, the optimizer will (Gelfand et al., 2000; Haario et al., 2005; Resnik et al., 2010):

- 1) find optimal values for the intercept, linear and quadratic terms. The optimization of this model with no "good" initial guesses for the intercept, and linear terms, is extremely demanding.

In this application, many parameters are optimized simultaneously. By shuffling values for all the parameters again in the quadratic term optimization step, we are disregarding all the work done in the previous step (linear model fitting) and cancel the principle of the use of the most likely values for the initialization of the algorithm. We understand the reviewer's concerns about the optimization in two steps. However, a fully quadratic term model with no initialization in "smart" (optimal intercepts and linear slopes) coordinates of the parametric space, for at least some of the model parameters, may yield inefficient estimates in converging due to overparameterization (Yehuda et al., 2021; Friedman et al., 2001, page 150 in model parameterization; Bates et al., 2015, a very interesting study with application in R language on how parsimonious modelling can assist in solving overparameterization issues independently of optimization (Bayesian or maximum likelihood) approach).

We have now rephrased the methods section in the manuscript and also added a better explanation of the quadratic terms addition to make the methods clearer (page 7).

Bergé, Laurent, Charles Bouveyron, and Stéphane Girard. "HDclassif: An R package for model-based clustering and discriminant analysis of high-dimensional data." *Journal of Statistical Software* 46.6 (2012): 1-29.

Lynch, Scott M. *Introduction to applied Bayesian statistics and estimation for social scientists*. Springer Science & Business Media, 2007.

Richardson, Sylvia, and Peter J. Green. "On Bayesian analysis of mixtures with an unknown number of components (with discussion)." *Journal of the Royal Statistical Society: series B (statistical methodology)* 59.4 (1997): 731-792.

Vats, Dootika, et al. "Analyzing MCMC output." *arXiv preprint arXiv:1907.11680* (2019).

Haario, Heikki, Eero Saksman, and Johanna Tamminen. "Componentwise adaptation for high dimensional MCMC." *Computational Statistics* 20.2 (2005): 265-273.

Gelfand, Alan E. "Gibbs sampling." *Journal of the American statistical Association* 95.452 (2000): 1300-1304.

Bărbos, A. C., et al. "Clone MCMC: parallel high-dimensional Gaussian Gibbs sampling." (2018).

Resnik, Philip, and Eric Hardisty. Gibbs sampling for the uninitiated. Maryland Univ College Park Inst for Advanced Computer Studies, 2010.

Dar, Yehuda, Vidya Muthukumar, and Richard G. Baraniuk. "A farewell to the bias-variance tradeoff? an overview of the theory of overparameterized machine learning." arXiv preprint arXiv:2109.02355 (2021).

Friedman, Jerome, Trevor Hastie, and Robert Tibshirani. The elements of statistical learning. Vol. 1. No. 10. New York: Springer series in statistics, 2001.

Bates, Douglas, et al. "Parsimonious mixed models." arXiv preprint arXiv:1506.04967 (2015).

Comment 8

- The condensing of five pathways into two can and should be supported by quantitative analysis. There is no statistical comparison problem if the authors simply compare the estimated model parameters at each time point across each of the five clusters to see how well the modelled brain maps match across different points of the trajectories across clusters.

Answer to comment 8

We agree with the reviewer that a quantitative justification of our claim will enforce the study results and overall message. The reviewer is correct, we reserved the numerical calculations behind the choice of pathways because we thought that the atrophy patterns, mean atrophy slopes and intercepts, demographics, and all other characteristics of the five clusters were enough to support the two-pathway conclusion. We have now changed the order of the figures and brought the heatmaps of random effects for each cluster (intercept and slopes) of patients in the main manuscript. **Figure 4** shows those heatmaps. Hierarchical clustering was applied posthoc in each cluster's intercepts and slopes separately, to identify the outlier clusters as well as the clusters that belong to a similar family. Intercepts help to understand baseline differences between the clusters, but slopes are the most important feature for us, since they provide information on longitudinal deterioration of gray matter. On the top of the figure for slopes one can see the clustered clusters in the form of a dendrogram. Minimal and limbic predominant atrophy, together with limbic predominant atrophy plus clusters are grouped together to give birth to the so called mediotemporal atrophy pathway, while hippocampal sparing together with the diffuse atrophy are grouped together to make the so called cortical atrophy pathway. However, one should interpret the addition of the diffuse atrophy pattern in the cortical pathway because the intercepts present this as an outlier cluster. We have added our approach for the pathway identification in the methods (see **page 9**) and in the results (see **pages 11-12**) sections.

Minor comments

Comment 9

Table S1 is useful in understanding the study data, however it doesn't show where these individuals fall in terms of time from AD onset. It would be beneficial to show a histogram or similar of time from AD onset vs. number of subjects to gain an idea of which portions of the trajectories are well sampled. It would be useful to also show this information after assigning individuals into clusters to check that the trajectories are well sampled in each cluster.

Answer to comment 9

We completely agree with the reviewer. This feature adds further information that adds to the characterization of the different atrophy patterns. We want to thank the reviewer for pointing this out. Indeed, these additional data shows that the more cortical atrophy a patient shows at the AD onset, the less MRI acquisitions are available for them in late disease stages (above ~70 months), which supports further the theory that hippocampal sparing/early onset AD patients have more aggressive disease course and drop out earlier than patients with more prototypical AD atrophy patterns at the AD onset. This observation may be seen as further support to the two pathways interpretation in our study (the more cortical atrophy at the AD onset, the higher is the chance of a faster/steeper disease course). A table has now been added in the supplementary material (**Table S9**) with the information asked by the reviewer (see results section, **page 12**).

Comment 10

In the methods the authors refer to atrophy levels, when I think they mean volumes. Atrophy implies a change in volume measured over time, but I think in this case cross-sectional volumes were input into a longitudinal model to assess atrophy over time and so the inputs are volumes, not atrophy levels.

Answer to comment 10

We apologize for the confusion. The original data used for this study were subcortical volumes and cortical thicknesses extracted by MRIs that were processed with the longitudinal pipeline of the Freesurfer software. The Freesurfer software has a longitudinal pipeline that takes repeated measurement MRIs from individuals as input and produces an individual specific template. Then Freesurfer processing based on the template gives outputs of longitudinal thickness and volume values (based on cross-sectional estimations). We agree with the reviewer that atrophy refers to reduction in thickness/volume over time. Based on the correction based on the cognitively unimpaired group of individuals, atrophy was observed in all the AD patients (all patient w-values were below zero). Therefore, we referred to atrophy and not thickness or volume after the clustering step. The reviewer is also correct that input data are in volume/thickness and not atrophy. **All this is now corrected throughout the manuscript.**

Comment 11

This sentence needs revising: "The word complex is used to showcase that probabilistic mixture modelling (number of clusters) is combined with mixed effect modelling (number of brain regions) in one model definition."

Answer to comment 11

We have now rephrased the sentence (see **page 6** of the manuscript, bottom paragraph).

Comment 12

The longitudinal clustering model would benefit from being specified mathematically.

Answer to comment 12

We agree with the reviewer. A mathematical formulation is a feature that we have also considered for this manuscript. However, we have already added a large amount of excessive text, figures and table (especially supplementary). We believe that an extra level of complexity in the manuscript might decrease the readership's interest, since we are focusing on the AD heterogeneity and not the methodology which was already presented in two recent publications (Poulakis, 2020-2021; see Komarek and colleagues (2014) for a simple mathematical description of the clustering model and Komarek and colleagues (2013), for the full mathematical description of the model including all the prior distributions and their starting values settings). References to other studies where the model was used are added on **page 6** of the manuscript, where the clustering method is described.

Poulakis, Konstantinos, et al. "Longitudinal deterioration of white-matter integrity: heterogeneity in the ageing population." *Brain communications* 3.1 (2021): fcaa238.

Poulakis, Konstantinos, et al. "Fully Bayesian longitudinal unsupervised learning for the assessment and visualization of AD heterogeneity and progression." *Aging (Albany NY)* 12.13 (2020): 12622.

Komárek, Arnošt, and Lenka Komárková. "Capabilities of R package mixAK for clustering based on multivariate continuous and discrete longitudinal data." *Journal of Statistical Software* 59.1 (2014): 1-38.

Komarek, Arnost, and Lenka Komárková. "Clustering for multivariate continuous and discrete longitudinal data." *The Annals of Applied Statistics* (2013): 177-200.

Comment 13

Why are field strength and eTIV included in the longitudinal clustering model, but age and cohort effects regressed out in the CU population? Why not do both at once?

Answer to comment 13

Thanks for this question. The focus on our modelling is in incorporating as much information as possible in the clustering step and not in preprocessing steps such as the w-value calculation. This enhances the ability of the clustering model to assess patterns in AD conditional to all the population (fixed effects) characteristics of the sample. However, two features needed to be assessed before clustering for practical reasons.

- Age was used in the step where age-related normal brain changes were subtracted from the AD data, hence it cannot be used again in the clustering as a fixed effect.

- Cohort was used as a random effect in the w-value calculation to assess the cohort effect in the healthy data before it interacts with disease (AD patient data). Moreover, the cohort effect was assessed before clustering as a random effect so that future data from other cohorts can be utilized for clustering based on our algorithm. Had it been added to the clustering model, the levels of cohorts in the population would have been fixed (only ADNI, J-ADNI, and AIBL), Thus, future clustering of new data (other cohorts) through the model would have been impossible. We have now added this information in the methods section (longitudinal clustering, 1st step, **page 6** of the manuscript).

Comment 14

The exact calculation for the “percentage of MCMCs with higher autocorrelation compared to the majority of MCMCs” needs writing down.

Answer to comment 14

We thank the reviewer for pointing this out. It is now written down under **supplementary table S3**.

Comment 15

The claim in the introduction that “the most urgent questions are ... if these subtypes finally converge at advanced stages of the disease” needs referencing given that the authors agree that it is inevitable. Otherwise this should be removed as a key insight of the study.

Answer to comment 15

In our opinion, it is an urgent question to support this claim with data although theoretically increasing brain atrophy should bring most of the AD patients at similar atrophy levels after a long disease duration. This is one of the three hypothetical outcome scenarios in AD heterogeneity, according to Ferreira and colleagues’ systematic review (Ferreira et al., 2020). We have now rephrased the discussion section to align with the reviewer’s suggestion (see **page 18**).

Ferreira, Daniel, Agneta Nordberg, and Eric Westman. "Biological subtypes of Alzheimer disease: A systematic review and meta-analysis." *Neurology* 94.10 (2020): 436-448.

Reviewer #3 (Remarks to the Author):

My comments below:

Comment 16

· I feel that the authors have responded appropriately. I appreciate their willingness to address areas where changes in writing style would strengthen the overall manuscript.

Answer to comment 16

We would like to thank the reviewer for the kind words. We believe that the **changes** in language in parts of the manuscript has increased its overall quality.

Comment 17

· The reference to Ferreira et al. appears to account for potential differences in comorbidities between AD subtypes. The authors’ acknowledgement of limited data with respect to comorbidities is reasonable and, in my opinion, the inclusion of an additional citation to this point adequately respects an area of study that is yet unexplored. The newly added supplementary table S9 is also quite comprehensive in terms of the body systems that it addresses in the ADNI and J-ADNI cohorts. Given the paper’s focus on neuroimaging, I do not believe that it is necessary at this time to pursue further in-depth analysis of non-neurologic disease burden.

Answer to comment 17

We completely agree with the reviewer that a further analysis of non-neurologic disease data falls off the aim of the

study.

Comment 18

- I appreciate the authors' elaboration of their use of graph theoretical approaches. In looking at the manuscript, I am satisfied by their statement that "nodal strength calculation was not used as the main analytical step in this study but only to help summarize the information from the cluster covariance matrices."
- I do think that the authors would do well to insert several sentences further delineating the difference between cross sectional atrophy correlations and longitudinal correlations. They do this well in their response document, wherein they state: "We did not want to suggest structural relation between brain regions just because of structural correlation, and this is why we focused mainly on the longitudinal correlation between brain regions"
- Given the importance of the time axis in the study's main conclusions, I feel that very clearly stating the essential differences in longitudinal vs. cross-sectional structural correlations will improve the Discussion section.

Answer to comment 18

Following the reviewer's suggestion, we have edited the discussion and added further notes on the difference between cross sectional and longitudinal correlations related to graph analysis (please see discussion, **page 17**).

Comment 19

- I believe that the authors' explanation of their use of nodal strength clarifies this issue, and the addition of additional material to pages 14 and 15 is appropriate. I feel that they have made sufficient adjustments to comment on pathological changes in the respective subtypes, and the role of nodal strength is clearer now. My only concern is that reviewers have cited a preprint on medRxiv here (which is not peer-reviewed).

Answer to comment 19

We agree with the reviewer that a peer reviewed study would increase the scientific certainty in our discussion. The paper of Vogel and colleagues is now published in Nature medicine. We have replaced the citation with the updated one (**reference number 10**).

Vogel, Jacob W., et al. "Four distinct trajectories of tau deposition identified in Alzheimer's disease." Nature Medicine 27.5 (2021): 871-881.

Comment 20

- The authors in their response provide a very intuitive and well-backed rationale for electing the 5-cluster solution over the 2-cluster solution. Based on their explanation, the 5-cluster solution provides a greater degree of resolution by which to probe structural changes in the brain whereas the 2-cluster solution stratifies severe vs. mild changes on a much coarser level. I do like this explanation quite a bit, however, I feel that it should be reflected in their manuscript. As it currently stands, they write "Since different spatial atrophy subtypes are more interesting from an exploratory perspective, we chose to interpret the results of the 5-cluster solution." I feel that this is insufficient relative to what they have provided in the response, which is clearer and has greater backing in evidence.
- I also find slight issue with the use of "interesting", as it could be seen as assuming an opinion on the part of a reader.

Answer to comment 20

We have changed the results section where the statement "more interesting" is made and we added a reference (**reference number 7**) to back our statement according to the reviewer's suggestion. We rephrased the word interesting in order to focus more on the numerical evaluation and deep phenotyping of atrophy subtypes that yielded from the 5-cluster solution compared to the 2-cluster solution (see **page 10**).

This is part of the answer to comments 1 from reviewer 1 and comment 4 from reviewer 2.

R simulated experiment

The simulated experiments in R language (text in blue is code and output and green is our comments) with comments is presented at the bottom of the document as R simulated experiment.

#This is an example of clustering only one variable.

#We will 1) simulate data, 2) apply clustering to them, 3) train a classifier with the cluster as label, 4) show that extreme values are classified by the model without errors but their distribution is completely different (the classifier is of low quality for generalization).

#simulate 80 values from two normal distributions. We call this variable age for the sake of simplicity in interpretation. The reviewer can see the data as 80 age realizations (sample) from a population.

```
myseed <- 500
set.seed(myseed)
options(scipen = 100)
age_variable <- c(rnorm(40,50,2),rnorm(40,75,5))
Original Age data
[1] 51.93698 53.93074 51.77265 50.06108 51.89911 48.84654 51.44305 51.23820 50.04201 50.54970 47.16473
46.43738 48.76073 51.01709 46.49411 47.01019
[17] 50.59835 49.37120 51.87226 52.94198 48.27840 51.93568 52.11050 50.38267 48.57904 48.53053 49.83714
48.26454 47.18250 53.59795 46.14366 52.02493
[33] 52.94789 49.43978 48.11785 49.16134 46.84588 52.23885 49.07103 47.82523 72.98267 84.08706 74.78149
79.15705 75.93913 76.35150 72.88693 73.21011
[49] 69.57854 72.45141 80.14777 78.02900 86.01858 71.85683 80.48500 76.76516 74.34665 65.07652 74.61503
66.74624 83.02239 69.06993 78.36534 64.69467
[65] 85.67421 73.06473 75.67391 69.13852 77.30644 71.98288 72.39803 79.30095 77.34529 64.49900 69.85473
82.76008 71.15869 78.25443 75.74990 70.35466
```

#plot the histogram of this variable and identify that the distribution of the data is bimodal

```
myhist <- hist(age_variable,breaks=20)
multiplier <- myhist$counts / myhist$density
mydensity <- density(age_variable,kernel="biweight",bw = "SJ")
mydensity$y <- mydensity$y * multiplier[1]
lines(mydensity,col=1)
```

#apply k means clustering to the data assuming that the ground truth does not exist, that is the data are unlabelled

```
my_simple_clustering <- kmeans(age_variable,centers=2)
```

#The clustering optimizer classified the data in two classes, we make a dataset with the continuous variable under consideration and the cluster number

training data cluster assignment

```
1 2 3 4 5 6 7 8 9 10 11 12 13 14 15 16 17 18 19 20 21 22 23 24 25 26 27 28 29 30 31 32 33 34 35 36 37 38 39 40 41
```

#simulate some data from distributions that were the model was not trained at. Tailored to our example, simulate data from much lower ages than 50 and much higher ages than 75 years.

```
glm.probs_unseen_extreme_data <- c(rnorm(10,10,2),rnorm(10,80,2))
```

new unseen data from different age distribution

```
[1] 9.379850 6.914852 9.122144 10.834764 11.128196 9.678910 11.313468 7.600630 12.617612 9.565516
78.793502 80.893083 82.210684 78.399349 81.618426
[16] 82.516148 80.264735 82.996753 81.514809 79.063784
```

```
glm.probs_unseen_data <- predict(my_classifier, newdata
```

=

```
data.frame("age_variable"=glm.probs_unseen_extreme_data), type = "response")
```

```
print(glm.probs_unseen_data)
```

classification of the new extreme unseen data

```
ifelse(glm.probs_unseen_data < 5, 1, 2)
```

```
1 2 3 4 5 6 7 8 9 10 11 12 13 14 15 16 17 18 19 20
```

```
1 1 1 1 1 1 1 1 1 1 2 2 2 2 2 2 2 2 2 2
```

```
new_hist <- hist(glm.probs_unseen_extreme_data, breaks=50)
```

#The classifier predicts allocations for the new unseen data based on the trained data (mean ages 50 and 75) with very high certainty, but the new unseen data are derived from different distributions (mean age 10 years old and 80 years old).

```
plot(myhist, xlim=c(0,100))
```

```
plot(new_hist, col=2, add=T, xaxt = 'n', yaxt = 'n')
```

```
legend(x = "topright", legend = c("training data", "new data"), fill = c(1, 2))
```

Histogram of age_variable

Reviewers' Comments:

Reviewer #2:

Remarks to the Author:

My outstanding concerns regard the validation, the inference of the five clusters being two pathways, and whether the reporting of results is balanced.

Validation:

The validation doesn't appear to be convincing; the following points need clarifying.

- In Figure S7 there are different numbers of clusters in each dataset and some clusters have very small numbers – in each dataset there are two clusters that contain three subjects or fewer.
- In J-ADNI + AIBL clusters 2 and 3 (the only ones with more than two subjects) look the same for the first 48 months from AD onset and only differ at 72 and 96 months, so it seems that they should be a single cluster.
- It is unclear whether the cluster numbers are intended to correspond in the two datasets, e.g. is cluster 1 in ADNI supposed to be similar to cluster 1 in J-ADNI + AIBL? Visually they don't look similar and a quantitative analysis of the similarity of the atrophy maps is still lacking.
- The characteristics of the clusters in Table S6 don't look similar and there is no statistical testing of the similarity of the characteristics between the clusters or compared to the whole cohort.
- The concordance in Table S7 is low.

Condensing five pathways into two pathways:

The logic of whether there are five clusters or two clusters and how many pathways this corresponds to is still unclear.

- In the results the authors state: "The 2-cluster solution (Figure S3) separated the discovery set only in terms of cortical severity (high versus low brain atrophy), whereas the 5-cluster solution revealed different spatial atrophy subtypes." but it isn't clear from which information this is inferred.
- The hierarchical clustering the authors provide in Figure 4 to support their inference of two pathways seems illogical – they choose the five-cluster solution over the two-cluster solution but then cluster the five-cluster solution to give two clusters.
- In general, the authors need to define what they mean by a "pathway". From the discussion it seems that they mean that at baseline individuals with AD belonging to each of the five clusters may be at different stages of the same overall progression pattern. Why not then test this directly by quantitatively comparing atrophy patterns at each stage of each progression pattern with all of the other stages of the other progression patterns?

Reporting of results:

There are a few areas of the manuscript where the reporting seems unbalanced.

- The abstract mentions that there are two pathways but doesn't mention that there are five clusters, which is the focus of the main manuscript.
- This statement in the Introduction is inaccurate: "However, the identified patterns may still reflect different disease stages since longitudinal information was not used. Assuming individuals will remain in the same subtype when using one observation and without longitudinal evidence is questionable.". To say that longitudinal information was not used and that there is no longitudinal evidence is incorrect as the studies validate their findings longitudinally.
- The authors don't mention in the limitations that there are no pre-AD scans included. The ideal dataset for modelling longitudinal trajectories of AD would look at scans from before and after an AD diagnosis in amyloid positive individuals. Only including scans after the onset of AD limits the portion of the trajectory that can be modelled to only the symptomatic phase, but it is well known that atrophy begins significantly before a diagnosis of AD. Therefore there is a possibility that some progression pathways are missed.

Reviewer #3:

Remarks to the Author:

The authors have address all my concerns and I do not have any further comments.

We would like to thank the reviewers for their feedback. Below you can find a point by point answer to the reviewers' comments, accompanied by the manuscript page numbers, where additional text has been added or adjusted for the needs of the revision.

REVIEWER COMMENTS

Reviewer #2 (Remarks to the Author):

Comment 1

My outstanding concerns regard the validation, the inference of the five clusters being two pathways, and whether the reporting of results is balanced.

Answer to comment 1

We thank the reviewer for helping us to address these points where our study results could become clearer. We have made extensive changes in our manuscript so that all concerns are answered adequately. For clarity, in this revised version of the manuscript we focused on discussing the five patterns of atrophy/neurodegeneration rather than the two pathways of atrophy/neurodegeneration. We only mention the two pathways of atrophy in the discussion where we summarize and interpret the results of the study.

Validation:

Comment 2

The validation doesn't appear to be convincing; the following points need clarifying.
- In Figure S7 there are different numbers of clusters in each dataset and some clusters have very small numbers – in each dataset there are two clusters that contain three subjects or fewer.

Answer to comment 2

We understand the reviewer's concern. Below we explain why we find different clustering solutions between ADNI and J-ADNI/AIBL and why some clusters have very few observations.

We start from the low number of patients in some clusters in each dataset. One of the features that are underseen in clustering regards the capability of a clustering method to identify clusters of certain sizes (or proportion to the overall sample). Our model choice is based on the clinical knowledge regarding AD subtypes. We already know that at the population level, less typical presentations of Alzheimer's disease (AD) have a low prevalence (see page 7 where we discuss about patient probabilities to belong in each cluster and page 18 where low prevalence clusters are discussed). Moreover, the chances to find representatives of those low prevalence subtypes in samples such as ADNI, AddNeuroMed, J-ADNI, and AIBL are lower because most of the patients enrolled in these studies often show a more typical phenotype (see discussion, page 16, 1st paragraph). Moreover, due to sampling biases in realistic cohort formation scenarios (such as in the case of ADNI, AddNeuroMed, J-ADNI, and AIBL), it can be very demanding to find numbers of patients from the population that correspond to the true (true here refers to the real percentage of the subtype in the whole AD population) estimated percentages of AD subtypes. This happens because of sampling uncertainty (sampling theory, already elaborated in the previous sentence) as well as the issue of misdiagnosis in some more atypical AD cases (Geidy et al., 2020). Thus, it is very likely that our datasets underrepresent the low prevalence subtypes.

Based on this background knowledge we chose a modelling approach that can discover clusters with very low sample representation if they exist (see added discussion about this on page 18). These can

be called outlier clusters in case they seem to present an outlier behavior, or likely evidence of heterogeneity in the dataset when we have information that can associate them with the previous literature. For example, k-means which is unquestionable one of the most popular clustering methods, has the tendency to produce convex shaped clusters (it tends to equalize the spatial variance of identified clusters) that in many instances become similar in size (Celebi, Kingravi, and Vela 2013). Therefore, k-means is a method that cannot easily identify AD subtypes because it drives the clusters to be approximately equal in size. Also, other models such as growth mixture modelling (Nilam and Grimm 2009), often constrain the minimum number of observations in each cluster, so that the results will be more generalizable and reproducible in external datasets. Such methods work very well in some diseases, where subtypes have equally large prevalence, but they suffer in diseases with rare subtypes (see method pages 5-7 and discussion page 18). Another issue that we discussed in the previous review round is that by splitting the overall sample to smaller datasets (split by cohort), the separate cohort analyses will return clusters with smaller numbers, especially in those atypical or outlier cases (see also Geidy et al., 2020). As a rule of thumb, our large experience with clustering has shown that the smaller a dataset is, the lower is the chance of identifying underrepresented clusters of patients due to reduced chances of population representativity in the sampling. This is a common issue in small sample analyses and relates to variability in variance estimates among other issues. When small samples are used the variance in those datasets (separate cohort analyses) tends to be miscalculated. For our application, this means that disease heterogeneity is observed less accurately in the separate cohorts than in a greater sample (full discovery dataset). This behavior is well discussed with simulated examples in the work of Baliga (2019) (see link below the answer). Moreover, the percentages of the clusters in the separate cohort analyses (Table S6) are not very different from the percentages of small clusters in the overall discovery dataset (Table 3), showing that the results of the main analysis are similar to the results in the separate cohorts.

Moving to the reviewer's concern regarding the different numbers of clusters in the two datasets, the different numbers of clusters are related to the characteristics of the datasets and also the nature of our clustering method (the distance between individual observations based on their characteristics). The clustering model fitted to ADNI (n=207) used a much greater sample than the J-ADNI/AIBL (n=90+23) model. Based on our methodological approach and given the previous observations (previous sentence), there are higher chances to find more clusters in ADNI than in J-ADNI/AIBL, because greater samples will reveal disease heterogeneity in the population more accurately (Baliga, 2019) and also because more patients will reveal more severity information since patients are sampled at different disease stages where atrophy severity varies (Ferreira and colleagues, 2020). However, this does not mean that there are more AD subtypes in the North American population (ADNI) compared with the Australian/Japanese populations (AIBL/J-ADNI). One thing that is important to mention here is that clustering by itself is a methodological approach where instability of results varies considerably when changing the population (Halkidi et al., 2001). Sometimes this happens even by excluding one or two observations from the sample. This is not a weakness of clustering but a strength of it, because it shows that it is an exploratory method that allows to assess different features of a population by adding or reducing the variability of it. To conclude, we expected different numbers of clusters in the two datasets. In our opinion, this finding would have been concerning only if there was a clear pattern (large representation) in one dataset but not in the other, because that would mean that different datasets include different subtypes of AD (see also page 18 of the discussion).

All in all, dividing the overall dataset in this study into two smaller datasets may be good for validation but one must be aware that this reduces the probability that each subset will be representative of the larger dataset (less variability in each subset), therefore reducing the chance of getting the exact same number of clusters and individuals within datasets (variance estimation issues discussed above) (Baliga., 2019).

We took the reviewer's comment as an opportunity to add a paragraph on page 18 of the discussion

to discuss the different numbers of clusters and the small clusters identified in the separate cohort analyses, in relation to clustering in small datasets and outlier/rare subtype identification.

References:

Serrano, Geidy E., et al. "Cardiac sympathetic denervation and synucleinopathy in Alzheimer's disease with brain Lewy body disease." *Brain communications* 2.1 (2020): fcaa004.

Celebi, M. Emre, Hassan A. Kingravi, and Patricio A. Vela. 2013. "A Comparative Study of Efficient Initialization Methods for the K-Means Clustering Algorithm." *Expert Systems with Applications* 40(1): 200–210. <https://linkinghub.elsevier.com/retrieve/pii/S0957417412008767>.

Ram, Nilam, and Kevin J. Grimm. "Methods and measures: Growth mixture modeling: A method for identifying differences in longitudinal change among unobserved groups." *International journal of behavioral development* 33.6 (2009): 565-576.

Vikram B. Baliga, <https://vbaliga.github.io/dangers-sample-variance-small-sample-size/>, (2019)

Ferreira, Daniel, Agneta Nordberg, and Eric Westman. "Biological subtypes of Alzheimer disease: A systematic review and meta-analysis." *Neurology* 94.10 (2020): 436-448.

Halkidi, Maria, Yannis Batistakis, and Michalis Vazirgiannis. "On clustering validation techniques." *Journal of intelligent information systems* 17.2 (2001): 107-145.

Comment 3

- In J-ADNI + AIBL clusters 2 and 3 (the only ones with more than two subjects) look the same for the first 48 months from AD onset and only differ at 72 and 96 months, so it seems that they should be a single cluster.

Answer to comment 3

We understand the reviewer's concern. In figure S7 clusters 2 and 3 look very similar. The main difference between them is the trajectories of cortical thinning and subcortical loss of volume in hippocampus and thalamus. These are clusters of individuals that show the same pattern of atrophy (spatial distribution), but they have different decline rates and starting times (related to AD clinical onset). We believe this information is very important and really unique since all previous studies in the field have modelled cross-sectional data as oppose to our current study based on longitudinal clustering. Further, the clusters were not formed solely based on the visualized fitted values that we see in figure S7, where we present only the fitted values below 1.6 standard deviations from the CU population. A more complete view of the differences between clusters 2 and 3 in the J-ADNI/AIBL dataset can be seen in figure S8. Cluster 2 compared to cluster 3 has more cortical thinning at the AD onset, but also prominent frontolateral and precuneus thickness differences at baseline and longitudinally.

These differences between clusters 2 and 3 of the J-ADNI/AIBL datasets, can be observed in their fitted values which are presented in the supplementary analysis at figure S8. We have also added a sentence (page 13, results) in the manuscript stating that the similarities can be found in the supplementary figure S8.

Comment 4

- It is unclear whether the cluster numbers are intended to correspond in the two datasets, e.g. is

cluster 1 in ADNI supposed to be similar to cluster 1 in J-ADNI + AIBL? Visually they don't look similar and a quantitative analysis of the similarity of the atrophy maps is still lacking.

Answer to comment 4

Regarding the patterns found in the separate cohort analyses, we aimed to make a comparison based on the fitted values of the complete discovery dataset and the separate datasets. However, as the reviewer points out, this may be insufficient. In this revised version, we did one more supplementary analysis to quantitatively compare the intercepts/slopes of all three clustering solutions. In this analysis, we first calculated the intercepts and slopes of the atrophy patterns of the overall model (model applied to the whole discovery dataset, 5 longitudinal patterns of atrophy), the ADNI model (5 longitudinal patterns of atrophy), and the J-ADNI/AIBL (4 longitudinal patterns of atrophy). This sums to 14 patterns of atrophy having 41 intercepts and 41 slopes. Then, intercepts and slopes were mean centered and unit scaled to have equal contribution in the distance metric. Finally, we calculated the distance between the 15 longitudinal atrophy patterns to assess which patterns from the separate analyses for ADNI and J-ADNI/AIBL match the five longitudinal patterns of atrophy that we found in the main analysis of our study.

The results were as follows:

- A. Patterns that match the most between the discovery set analysis and the separate cohort analyses:
 - a. MA similar to ADNI cluster 3 and J-ADNI/AIBL cluster 3
 - b. LPA similar to ADNI cluster 2 and J-ADNI/AIBL cluster 2
 - c. LPA+ similar to ADNI cluster 1 and J-ADNI/AIBL cluster 3
 - d. DA similar to ADNI clusters 4 and J-ADNI/AIBL cluster 4
 - e. HS similar to ADNI cluster 2 and J-ADNI/AIBL cluster 1
- B. Patterns that match the most between ADNI and J-ADNI/AIBL cohorts:
 - a. ADNI cluster 1 and J-ADNI/AIBL cluster 3
 - b. ADNI cluster 2 and J-ADNI/AIBL cluster 2
 - c. ADNI cluster 3 and J-ADNI/AIBL cluster 2
 - d. ADNI cluster 4 and J-ADNI/AIBL cluster 4
 - e. ADNI cluster 5 and J-ADNI/AIBL cluster 2
 - f. J-ADNI/AIBL cluster 1 and ADNI cluster 1 and 3 (based on its fitted values, J-ADNI/AIBL cluster 1 presents an HS pattern as well. In Figure S7, where fitted values are presented, it looks similar to the HS longitudinal pattern of atrophy (figure 2 of the main manuscript too).

However, clusters 4 and 5 of ADNI and clusters 1 and 4 of J-ADNI/AIBL are clusters of very low prevalence. The atrophy patterns discovered through the separate analyses are similar but not exactly the same to the ones discovered in the full dataset. This can be attributed to the application of the model in rather small datasets (discussed now on page 18 of the discussion). We have also elaborated on this topic in the answer to comment 2.

Thus, the results of the comparison between clustering solutions should be interpreted cautiously considering the small sample analysis and the measures used in the comparison that cannot capture the full complexity of the longitudinal clustering model.

We have now added this complementary analysis in the methods (page 8), results (page 13, results, model validation section), discussion (page 18) sections, and figure S8 legend.

Comment 5

- The characteristics of the clusters in Table S6 don't look similar and there is no statistical testing of the similarity of the characteristics between the clusters or compared to the whole cohort.

Answer to comment 5

We thank the reviewer for this suggestion. Statistical analysis for the assessment of within cohort and between cluster differences is reported in Table S6. These results strengthen our conclusions and they are now added.

Apart from the significant results reported in table S6, some observations that we made were: ADNI clusters 2 and 3 had close to significant differences in terms of MMSE ($p = 0.07355$), while clusters 1 and 3 had close to significant differences in CDR ($p = 0.09558$). Moreover, it is evident from Table S6 that clusters 4 and 5 which were not assessed due to low numbers have different years of education (more), Age at scan (lower), age at the onset of AD (lower), MMSE (worse), and CDR (worse) than clusters 1, 2, and 3. Moving to J-ADNI/AIBL, clusters 1 and 4 that also had low numbers and were not statistically contrasted to clusters 2 and 3, had qualitatively lower age at scan and age at the onset of AD, and worse MMSE, and CDR sum of boxes. All these results are listed in Table S6 and can be reached by the readership in case they are interested in the different cohort analyses. Please find the new results in Table S6 of the supplementary material. Explanations regarding which groups were statistically contrasted are in the legend of table S6.

Comment 6

- The concordance in Table S7 is low.

Answer to comment 6

We agree with the reviewer that the concordance seems low in some of the columns. This happens because of subsampling due to the split of the discovery dataset in two segments. We make this clear as a limitation in the discussion (page 18), especially related to the hippocampal sparing and diffuse atrophy patterns.

Comment 7

Condensing five pathways into two pathways:

The logic of whether there are five clusters or two clusters and how many pathways this corresponds to is still unclear.

Answer to comment 7

We understand the reviewer's concern. For clarity we now only refer to clusters throughout the abstract, introduction, methods, and results. As described in the results section we chose the 5-cluster solution over the 2-cluster solution (see page 10), see also comment 8.

We must make clear the path of our analysis here to understand the distinction between pathway analysis and cluster analysis. The overall clustering procedure was split in three steps: correction of the data for normal aging, clustering (with Bayesian mixed effect gaussian mixture modelling), and finally evaluation of the model in new unseen data (**also** evaluation in separate cohorts which is something that was added during the review process). When clustering was finished and the 5-cluster solution was chosen and interpreted, a further methodological step aided in understanding if a) the five clusters represent different stages of the disease, or b) some clusters represent the same spatial atrophy pattern over time, but with different starting point (interpretation anchored to AD clinical onset), and/or different atrophy deterioration rates. To do this we applied hierarchical clustering on the intercepts and slopes of the five discovered clusters. Through this approach we identified the two main atrophy patterns that we discussed under the name pathway in our manuscript. In the answer to

comment 9 we present the atrophy patterns of the 2-cluster solution that were not further interpreted as well as the 5-cluster solution that was used. These are the main findings that are presented in our manuscript. In the answer to comment 9 we also present the two pathways identified within the 5-cluster solution.

We now only refer to the pathways in the discussion section of the manuscript, where we interpret the 5-cluster solution. On page 14 (see below) we introduce the concept of pathways. We there condense the 5-clusters into two pathways (not to be confused with choosing the 5-cluster solution over the 2-cluster solution). Each of the two pathways include the clusters (from the 5-cluster solution) with similar atrophy patterns over time (however different starting points at clinical onset and/or rate of atrophy as well as clinical progression). Please see below an extraction from our revised manuscript:

“At the modelled clinical disease onset, our method successfully identified the same patterns of atrophy previously identified in neuropathological and neuroimaging subtyping studies (minimal atrophy, limbic predominant, typical AD, and hippocampal sparing). Our results revealed two main pathways of atrophy. We introduce the term pathway to describe AD patients that show similar spatial distribution of atrophied brain regions over time. Within the same atrophy pathway, patients may progress faster (LPA+) than others (LPA and MA) but their spatial distribution of atrophy over time is similar. This pathway contrasts with the second different atrophy pathway in AD, which has a different spatial distribution with mainly cortical atrophy over time. The differences in progression rates also reflect the rates of cognitive decline of the patients. It is a very important future aim to understand the factors underlying of these differences in progression within the same pathway but also between the different pathways that we have identified.”

Comment 8

- In the results the authors state: “The 2-cluster solution (Figure S3) separated the discovery set only in terms of cortical severity (high versus low brain atrophy), whereas the 5-cluster solution revealed different spatial atrophy subtypes.” but it isn’t clear from which information this is inferred.

Answer to comment 8

We agree with the reviewer that this is something that needs clarification. The 2-cluster solution (figure S3) shows that: 1) there is one cluster (cluster 1) that includes AD patients with no observable atrophy at the AD onset compared to the CU group, which slowly spreads all over the cortex, and 2) there is another cluster of patients (cluster 2) with prominent atrophy already at the AD onset, which spreads very steeply throughout the cortex. On the other hand, figure 2 (5-cluster solution) of the main manuscript shows a much more complex representation of atrophy patterns, that does not only show patterns based on severity as figure S3, but it also identifies different spatial patterns of atrophy (hippocampal sparing (mostly cortical) vs limbic predominant (with different cortical thinning trajectories over time)).

We have now clarified this in the results (see results, clustering evaluation, page 10) about the lack of further investigation of the 2-cluster solution yielded from the comparison between the fitted values presented in figure 2 and figure S3.

Comment 9

- The hierarchical clustering the authors provide in Figure 4 to support their inference of two pathways seems illogical – they choose the five-cluster solution over the two-cluster solution but then cluster the five-cluster solution to give two clusters.

Answer to comment 9

We understand the reviewer's concern. We do not go back to a 2-cluster solution, we summarize the 5-cluster solution in two pathways (similar spatial atrophy spread). We therefore would like to clarify that:

The 2-clusters solution includes:

- A. One cluster that shows minimal atrophy at the clinical AD onset. This atrophy extends slowly to the temporal lobe, medially and then laterally to reach the rest of the cortex later.
- B. Another cluster that shows wide-spread temporal, frontal, and parietal atrophy at the clinical AD onset. This cluster of patients shows very fast atrophy progression all over the grey matter.

The 5-cluster solution includes:

- A. MA, which is a cluster that shows minimal atrophy at the clinical AD onset. This atrophy extends slowly to the temporal lobe, medially and then laterally to reach the rest of the cortex later.
- B. LPA, which shows entorhinal cortex atrophy at the clinical onset, with later involvement of other temporal lobe regions including the hippocampus.
- C. LPA+, which shows atrophy spatially similar to LPA but exhibits more atrophy in the entorhinal cortex at AD onset. Atrophy progressively extended to the temporal lobe and then further to the rest of the cortex.
- D. DA cluster that shows increased temporal, frontal, and parietal atrophy at the clinical AD onset. This cluster of patients shows very fast atrophy progression all over the grey matter.
- E. HS, which shows parietal lobe atrophy and no involvement of medial-temporal lobe structures at disease onset, but fast atrophy progression.

The 2 pathways are the result of a quantitative interpretation of the 5-cluster solution and is obtained posthoc (see page 9 method):

- A. An atrophy pathway that shows the common spatiotemporal atrophy distribution of clusters A, B, and C (And partially D) of the 5-cluster solution.
- B. An atrophy pathway that shows the spatiotemporal atrophy distribution of cluster E (and partially D) of the 5-cluster solution.

To clarify, when clustering was finished and the 5-cluster solution was chosen and interpreted, a further methodological step aided in understanding if a) the five clusters represent different stages of the disease, or b) some clusters represent the same spatial atrophy pattern over time, but with different starting point (interpretation anchored to AD clinical onset), and/or different atrophy deterioration rates. To do this we applied hierarchical clustering on the intercepts and slopes of the five discovered clusters. Through this approach we identified the two main degeneration patterns that we discussed under the name pathway in our manuscript. In the answer to comment 9 we present the 2-cluster solution's atrophy patterns that were not further interpreted, the 5-cluster solution that was used and its main findings are presented in our manuscript, and the two pathways identified within the 5-cluster solution.

The fitted values in figure 2 (5-cluster solution longitudinal atrophy patterns) and the quantitative validation of those fitted values (hierarchical clustering, see previous paragraph) that can be seen on top of figure 4 in the form of a dendrogram provide us with the evidence that points out the existence of two atrophy pathways, described in the legend of Figure 4. By atrophy pathways we do not mean clusters of patients. We stress this here because an atrophy pathway includes AD patients that show similar spatial spreading of atrophy. However, within an atrophy pathway, a patient may develop an atrophy pattern and cognitive deterioration much faster than another patient, while within a longitudinal atrophy cluster, all patients should develop similar atrophy pattern with the same

progression rate after the clinical AD onset. The definition of pathways of atrophy that we have tried to use in our manuscript is comparable to Braak staging for tau (Braak and colleagues, 2006), a description of the stereotypical distribution of neurofibrillary tangles spreading in the AD brain, but with atrophy instead of tau. Whitwell and colleagues described the atrophy pathway that correlates to the Braak staging in 2008 (Whitwell et al., 2008)

For clarity we now only refer to clusters throughout the abstract, introduction, methods, and results. As described in the results section we chose the 5-cluster solution over the 2-cluster solution (page 10).

We now also only refer to the pathways in the discussion section of the manuscript, where we interpret the 5-cluster solution. On page 14 we introduced the concept of pathways while on page 9 we explained the quantitative method that yielded to them. We there condense the 5-clusters to two pathways. As stated before, each of the two pathways include the clusters with similar spatial atrophy distribution over time (however different starting points at clinical onset and/or rate of atrophy as well as clinical progression).

References:

Braak, H, et al. "Staging of Alzheimer disease-associated neurofibrillary pathology using paraffin sections and immunocytochemistry." *Acta neuropathologica* 112.4 (2006): 389-404.

Whitwell, J. L., et al. "MRI correlates of neurofibrillary tangle pathology at autopsy: a voxel-based morphometry study." *Neurology* 71.10 (2008): 743-749.

Comment 10

- In general, the authors need to define what they mean by a "pathway". From the discussion it seems that they mean that at baseline individuals with AD belonging to each of the five clusters may be at different stages of the same overall progression pattern. Why not then test this directly by quantitatively comparing atrophy patterns at each stage of each progression pattern with all of the other stages of the other progression patterns?

Answer to comment 10

We apologize that we had not made this clear enough. On page 14 we introduce the concept of pathways. Each of the two pathways include the clusters with similar spatial atrophy distribution staging in the brain:

"At the modelled clinical disease onset, our method successfully identified the same patterns of atrophy previously identified in neuropathological and neuroimaging subtyping studies (minimal atrophy, limbic predominant, typical AD, and hippocampal sparing). Our results revealed two main pathways of atrophy. We introduce the term pathway to describe AD patients that show similar spatial distribution of atrophied brain regions over time. Within the same atrophy pathway, patients may progress faster (LPA+) than others (LPA and MA) but their spatial distribution of atrophy over time is similar. This pathway contrasts with the second different atrophy pathway in AD, which has a different spatial distribution with mainly cortical atrophy over time. The differences in progression rates also reflect the rates of cognitive decline of the patients. It is a very important future aim to understand the factors underlying of these differences in progression within the same pathway but also between the different pathways that we have identified."

In the answer to comment 7 (see also page 9 in the revised manuscript for the method that was used to find the pathways), we also explained exactly how we compared (hierarchical clustering) the five different clusters of patients that expressed the five different longitudinal patterns of atrophy and finally summarized our results from the perspective of two pathways.

We believe that this further hierarchical clustering and interpretation into two pathways is needed and really is a step forward in the field. Many previous studies have converged into the existence of 3-4 clusters, but there is a constant question whether these clusters really represent distinct subtypes or rather, patients at different stages of the disease. Our current study is the first time that clusters are identified on longitudinal data and therefore, the first possibility in the literature to start understanding the question about distinct subtypes vs. disease stages. What we observe in our data is that the five different longitudinal clusters in our study may reflect aspects of both subtypes and stages. But with the question of subtype vs. stages in mind, our data suggest that the five clusters may indeed represent only 2 atrophy pathways. One pathway is primarily cortical and the other one is primarily medial temporal. This is the first time this is empirically observed, while some previous data from our lab and collaborators had already suggested these two pathways (Levin et al, 2021; Ferreira et al 2017, etc). We have now clarified our idea about pathways, and restricted that section to the discussion section. While we have now reduced discussion about pathways substantially, we believe this is one of the novel points of the current paper and would suggest keeping this shortened version in the discussion, which could also stimulate future work by the scientific community.

References:

Levin, Fedor, et al. "Data-driven FDG-PET subtypes of Alzheimer's disease-related neurodegeneration." *Alzheimer's research & therapy* 13.1 (2021): 1-14.

Ferreira, Daniel, et al. "Distinct subtypes of Alzheimer's disease based on patterns of brain atrophy: longitudinal trajectories and clinical applications." *Scientific reports* 7.1 (2017): 1-13.

Reporting of results:

There are a few areas of the manuscript where the reporting seems unbalanced.

Comment 11

- The abstract mentions that there are two pathways but doesn't mention that there are five clusters, which is the focus of the main manuscript.

Answer to comment 11

We agree, we have now improved the abstract. We now mention that the main finding of the study is the five longitudinal AD atrophy patterns. Moreover, we avoided the word pathway in the abstract, since we now use it only in a section of the discussion.

Comment 12

- This statement in the Introduction is inaccurate: "However, the identified patterns may still reflect different disease stages since longitudinal information was not used. Assuming individuals will remain in the same subtype when using one observation and without longitudinal evidence is questionable.". To say that longitudinal information was not used and that there is no longitudinal evidence is incorrect as the studies validate their findings longitudinally.

Answer to comment 12

Our previous statement was indeed inaccurate, we have now changed it to the following "However, we cannot exclude the chance that the identified patterns may still reflect different disease stages, since longitudinal information was not used for clustering, only for characterizing subtypes post hoc".

Comment 13

- The authors don't mention in the limitations that there are no pre-AD scans included. The ideal dataset for modelling longitudinal trajectories of AD would look at scans from before and after an AD diagnosis in amyloid positive individuals. Only including scans after the onset of AD limits the portion of the trajectory that can be modelled to only the symptomatic phase, but it is well known that atrophy begins significantly before a diagnosis of AD. Therefore, there is a possibility that some progression pathways are missed.

Answer to comment 13

We have now added in the limitation section a sentence to stress the absence of pre-AD scans to the (see limitations section in the discussion, page 18-19):

“Pre-AD scans were not included. This reduced our ability to infer atrophy patterns that precede the diagnosis of AD dementia.”

Reviewer #3 (Remarks to the Author):

Comment 14

The authors have address all my concerns and I do not have any further comments.

We thank the reviewer for the valuable feedback.

Reviewers' Comments:

Reviewer #2:

Remarks to the Author:

I am satisfied by the responses and changes made and have no further comments.